# Development of chemokine network inhibitors using combinatorial saturation mutagenesis
Jhanna Kryukova ⓘ , Serena Vales ⓘ , Megan Payne ⓘ , Gintare Smagurauskaite,
Soumyanetra Chandra ⓘ , Charlie J. Clark, Graham Davies & Shoumo Bhattacharya ⓘ ✉

Targeting chemokine-driven inflammation has been elusive due to redundant pathways constituting chemokine-immune cell networks. Tick evasins overcome redundant pathways by broadly targeting either CC or CXC-chemokine classes. Recently identified evasin-derived peptides inhibiting both chemokine classes provide a starting point for developing agents with enhanced potency and breadth of action. Structure-guided and affinity maturation approaches to achieve this are unsuitable when multiple targets are concerned. Here we develop a combinatorial saturation mutagenesis optimisation strategy (CoSMOS). This identifies a combinatorially mutated evasin-derived peptide with significantly enhanced $pIC_{50}$ against three different inflammatory disease chemokine pools. Using AlphaFold 3 to model peptide - chemokine interactions, we show that the combinatorially mutated peptide has increased total and hydrophobic inter-chain bonding via tryptophan residues and is predicted to sterically hinder chemokine interactions required for immune cell migration. We suggest that CoSMOS-generated promiscuous binding activities could target disease networks where structurally related proteins drive redundant signalling pathways.

The chemokine network in humans consists of 46 structurally related secreted proteins which signal through G-protein coupled chemokine receptors present on leucocytes[1]. Chemokine binding to cognate receptors activates directed migration of leucocytes i.e., chemotaxis, to the site of chemokine expression[1]. This network plays a key role in the recruitment of diverse leucocyte subclasses including neutrophils, monocyte/macrophages, T-cells and dendritic cells, and in immune-inflammatory diseases including atherosclerosis, type 1 diabetes, rheumatoid arthritis, and inflammatory bowel disease[2–5]. Based on the spacing of their N-terminal cysteine residues, chemokines are grouped into CC, CXC, CX3C and XC classes[1]. In humans, CC-chemokines, the largest class, have 26 members, while CXC chemokines have 17 members[1]. Chemokines may be grouped into inflammatory or homeostatic types, with inflammatory chemokines functioning in leucocyte recruitment to the sites of inflammation, and constitutively expressed homeostatic chemokines functioning in immune cell homeostasis[1]. Several chemokines have dual inflammatory and homeostatic roles[1,6]. Chemokines have a conserved tertiary structure, consisting of an unstructured N-terminus, an N-loop, a triple-stranded β-sheet connected by 30s and 40s loops, and an α-helix[7]. The chemokine N-terminus interacts with the transmembrane pocket of the receptor. The N and 40s loops and β1-strands contact the N-terminus of the receptor[7]. Interactions of chemokines with

other chemokines also have functional consequences. CXC-type heterodimers form by β1-strand interactions inhibit function, whereas CC-type heterodimers, formed by interactions between CC motifs and N-terminal residues enhance chemokine activity[7,8].

The chemokine network is characterised by redundant pathways that connect chemokines to immune cells[9]. This is created, at least in part, by the expression of multiple chemokines in inflamed tissues[9], "one-to-many" interactions between chemokines and receptors[10], and expression of multiple chemokine receptors by each leucocyte subclass[11–13]. Network redundancy results in robustness to environmental variation[14,15], leading to the failure of therapeutics that target individual chemokines or receptors in immuno-inflammatory disease[4,16].

Ticks overcome redundant pathways in chemokine networks by producing two structurally distinct classes of salivary proteins called evasins[17]. Class A and class B evasins bind and neutralize multiple CC-class and CXC-class chemokines respectively[17]. Binding occurs in a "one-to-many" manner, with the multi-targeting effect suppressing local inflammation[17–25]. This allows ticks to suppress inflammation at the bite site and feed for prolonged periods despite local expression of multiple chemokines[26,27]. We[21–24] and others[19,20] have cloned and characterised over 50 anti-inflammatory tick salivary evasins. Despite showing efficacy in multiple pre-clinical disease

Centre for Human Genetics and RDM Cardiovascular Medicine, University of Oxford, Roosevelt Drive, Oxford, OX3 7BN, UK.
✉e-mail: shoumo.bhattacharya@cardiov.ox.ac.uk

models[19,28–35], evasins are not attractive for direct clinical translation owing to their binding specificities requiring the use of both CC- *and* CXC-binding evasins, and potential immunogenicity[36].

We recently showed that 16-mer peptides from class A evasins, isolated by phage-display performed against multiple chemokines followed by deep sequencing, have the surprising ability to bind and inhibit chemokines from *both* CC and CXC classes[37]. Our analyses suggested that these peptides occlude different parts of the chemokine receptor-binding sites and contain two motifs, one with an anionic patch primarily contributing ionic interactions, and the second contributing hydrophobic interactions[37]. The lead peptide HD2 bound and inhibited CC-chemokines, but unexpectedly, also bound and inhibited two CXC-chemokines[37]. Our study suggested that HD2 could serve as a template for designing broad-spectrum chemokine inhibiting agents for inflammatory disease[37].

Approaches used to enhance the functional characteristics of peptides and proteins include affinity maturation, iterative saturation mutagenesis, and deep mutational scanning[38–40]. High-throughput approaches that shorten the discovery process include yeast display, phage-display, mRNA-display, and affinity-selection-mass-spectrometry[41–47]. These methods rely upon identification of peptides that bind the target with high affinity, which is a determinant of potency. Identification of combinations of improving point mutations is particularly challenging. It is known that combining different "improving" mutations identified by saturation mutagenesis can result in additive or co-operative effects[48,49]. However, combinatorial additive or co-operative effects can be difficult to predict, especially when structural information regarding target binding mechanism is lacking. From the perspective of developing inhibitors of *multiple* chemokines, limitations of the above methods are that they are typically applied to a *single* target, which may limit their use for identifying peptides that can bind multiple targets.

Our goal was to develop a method that would allow us to rapidly identify improving mutations in the broad-spectrum CC and CXC-binding evasin-derived peptides, but without relying on structural information or knowledge of binding mechanism. We developed a two-step method that initially applied saturation mutagenesis phage-display and screening with multiple chemokines to identify point mutations that improved binding. This was followed by screening of a phage-display library that combined all possible improving mutations. We term our method CoSMOS for

"combinatorial saturation mutagenesis optimization strategy". Using this method, we identify peptides that have significantly enhanced breadth of binding and inhibitory activity compared to parental and point mutant peptides. Our results indicate that CoSMOS can be used to generate the promiscuous binding activities required to effectively target disease networks where redundancy is generated by structurally related proteins.

## Results

### NNK mutagenesis library construction and screening
Our previous studies had indicated that the exemplar peptide HD2, despite originating from the CC-specific class A evasin EVA4, has the ability to bind 21 chemokines from CC and CXC/X3C (referred to as "CXnC") classes in phage display, and could significantly inhibit cell migration in response to CCL3, CCL5, CCL7, CCL8, CXCL6, and CXCL10[37]. To identify mutations that improved breadth and affinity of binding we generated a library of HD2 mutations where codons encoding each residue were substituted with NNK, to allow the incorporation of 20 canonical amino acids at each site. Using this library, we performed phage-display selection against a panel of 22 biotinylated chemokines, in parallel (Fig. 1a). These chemokines were chosen as they were commercially available. We used biotinylated complement C5A to identify and exclude non-specific binders as it resembles chemokines in molecular weight, extracellular location, GPCR binding, and chemotactic function[50]. We expressed mutant peptide enrichment (E) in phage-display in comparison to the input library as log2E, which is positively correlated with binding affinity[43]. For each mutant - chemokine pair we calculated Δlog2E as the difference in log2E compared to the parental HD2 peptide - chemokine pair (Fig. 1b).

### Identification of point mutations that improve chemokine binding
We used a two-pronged strategy to identify individual peptides with point mutations that improve chemokine binding over the parental peptide HD2. To identify mutations that improve overall chemokine binding affinity, for each mutant we determined the mean Δlog2E averaged over all chemokines. To include mutations that improve chemokine binding diversity, for each mutant we determined its peak Δlog2E. Using a ranking strategy based on both highest mean and peak Δlog2E across all chemokines, and across the CC and CXnC chemokine classes separately, we identified 16 "improving"

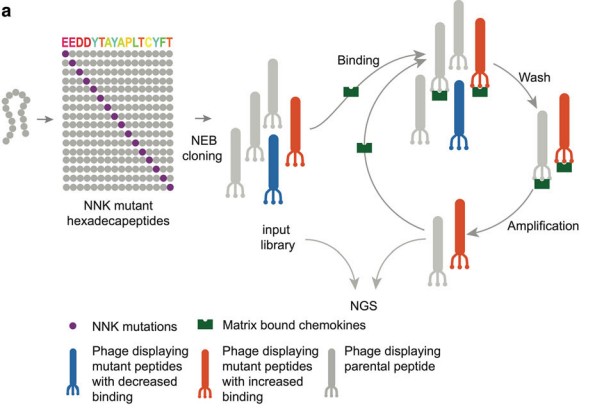

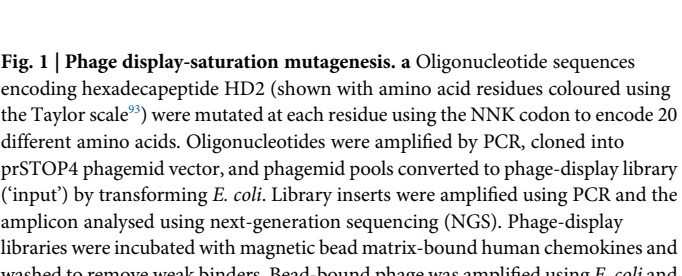

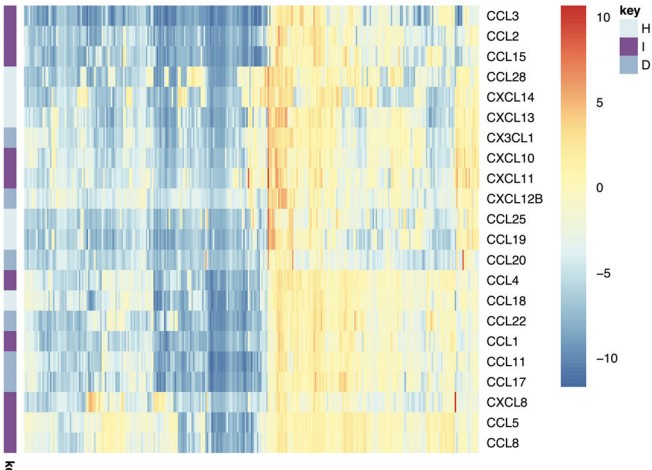

**Fig. 1 | Phage display-saturation mutagenesis. a** Oligonucleotide sequences encoding hexadecapeptide HD2 (shown with amino acid residues coloured using the Taylor scale[93]) were mutated at each residue using the NNK codon to encode 20 different amino acids. Oligonucleotides were amplified by PCR, cloned into prSTOP4 phagemid vector, and phagemid pools converted to phage-display library ('input') by transforming *E. coli*. Library inserts were amplified using PCR and the amplicon analysed using next-generation sequencing (NGS). Phage-display libraries were incubated with magnetic bead matrix-bound human chemokines and washed to remove weak binders. Bead-bound phage was amplified using *E. coli* and

helper phage, and the amplified library incubated again with matrix-bound chemokine to purify stronger binders. After three selection rounds the phage-inserts were amplified using PCR and amplicon analysed by NGS. Enrichment (E, expressed as log2E) following selection was calculated as ratio of output to input peptide frequencies. **b** Clustered heatmap showing impact of single mutations on Δlog2E compared to parental HD2 for all chemokines studied. Chemokine types in rows are indicated as I = inflammatory, D = dual, H = homeostatic. Colour scale bar indicates Δlog2E compared to parental peptide HD2 which is present in the library.

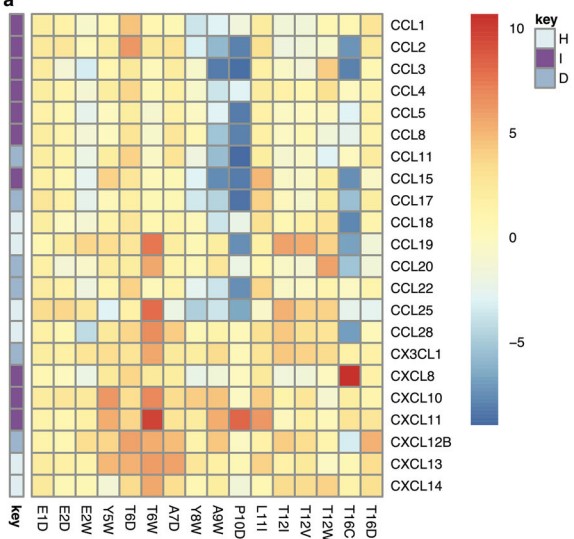

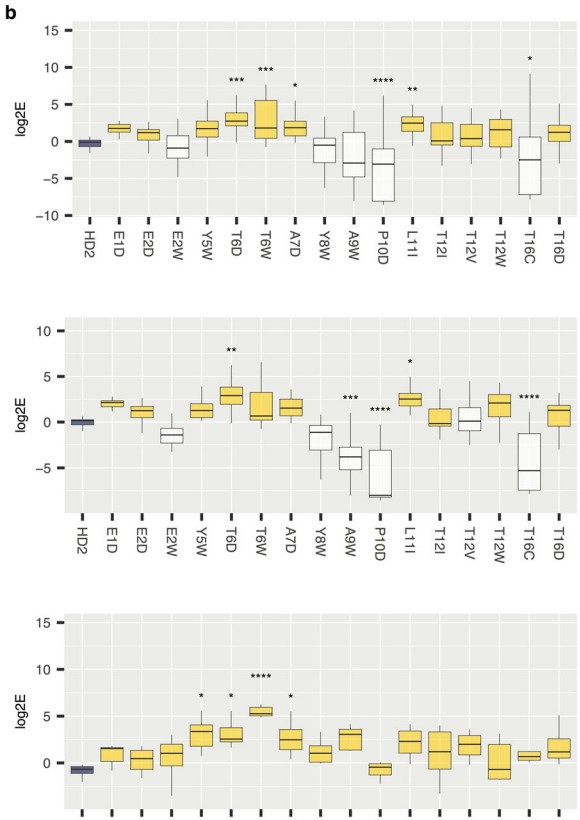

**Fig. 2 | Effect of HD2 mutations on chemokine binding in phage-display. a** Tile-plot showing impact of selected HD2 mutations on Δlog2E for the indicated chemokines. Chemokine types in rows are indicated as I = inflammatory, D = dual, H = homeostatic. Colour scale bar indicates Δlog2E. **b–d** Box-whisker plots showing impact of selected HD2 mutations (X-axis) on log2E (Y-axis) considered for all chemokines, for CC chemokines and for CXnC chemokines respectively. Each plot shows the median as centre, 25th and 75th percentile as bounds, and 1.5*interquartile range as whiskers. Statistically significant differences using a two-sided Dunnett's test with correction for multiple comparisons, are indicated by asterisks: * $P \leq 0.05$, ** $P \leq 0.01$, ***, $P \leq 0.001$, **** $P \leq 0.0001$. The control box in each panel is coloured blue, while boxes showing a positive value for difference from control >0.55 (identified from Dunnett's test) shown as yellow. Number of biologically independent experiments per group and exact P-values are provided in Supplementary Table 9a.

mutations at 11 different residues (Fig. 2a). Several mutations (e.g. T6D, T6W, A7D and L11I) significantly improved log2E compared to parental HD2 when considered for all chemokines (Fig. 2b). When CC chemokines were considered separately, mutations T6D, and L11I showed significant improvement, and when CXnC chemokines were considered separately, mutations Y5W, T6D, T6W and A7D showed significant improvement in log2E (Fig. 2c, d). Notably, the log2E for the parental HD2 was ~0 in these analyses, indicating lack of enrichment. The most likely explanation of this result is that mutant peptides with enhanced binding out-compete the parental HD2 peptide.

### Effect of point mutations on chemotaxis

We next evaluated the effect of parental HD2 and selected HD2 mutant peptides on chemotaxis of THP1[51] cells, activated CD8+T-cells (ATC), and a Jurkat cell line expressing CXCR1 (J:CXCR1), induced by seven CC and three CXC chemokines (Fig. 3a, Supplementary Figs. 1-3). We selected these cells as models of monocytes (THP1), T-cells (ATC) and granulocytes (J:CXCR1). We chose five chemokines (CCL5, CCL7, CCL8, CCL14, and CCL23) for their ability to induce migration of THP1 cells[37]. We chose four chemokines (CCL19, CCL21, CXCL9, CXCL11) for their ability to induce T-cell migration[52,53]. CXCL6, a granulocyte chemoattractant, was chosen for its ability to induce J:CXCR1 migration[37,54]. In these experiments we used the parental peptide HD2 as control to assess if the mutation had a significant impact on chemotaxis. The selected mutants retained the ability to inhibit THP1 cell migration induced by CCL5, CCL7 and CCL8 previously

observed with HD2 (Supplementary Fig. 1). Several mutants (e.g. Y5W, T6W, A9W, L11I, T16C) showed significant improvement in inhibition of CCL19, CCL21, CXCL9, and CXCL11-induced migration of ATCs compared to the parental HD2 peptide (Supplementary Fig. 2). Mutants T6W, A9W, and T16C showed significant improvement in inhibition of CXCL6-induced migration of J:CXCR1 cells compared to the parental HD2 peptide (Supplementary Fig. 3). We performed a meta-analysis of these results to evaluate overall impact over a range of cell migration assays (Fig. 3). The most effective individual mutations with significant improvement over HD2 in this analysis were Y5W, T6W, A9W, and T16C.

### Combinatorial mutagenesis library construction and screening

The above results indicate that it is possible to improve the chemokine-binding and inhibiting properties of an evasin-derived peptide using saturation mutagenesis and selection with multiple chemokines. To identify point mutations that have additive or co-operative effects when combined in a single peptide, we constructed a library encoding all possible (i.e., 3585) combinations of the 16 "improving" mutations identified by NNK-mutagenesis, phage-display and the prioritisation approach described above (Fig. 4a). The library contained in addition the parental HD2, and the 16 single mutants. We performed phage-display selection of the combinatorial library against a panel of 25 commercially available biotinylated chemokines, and calculated log2E and Δlog2E for each mutant - chemokine pair in relation to the parental HD2 peptide - chemokine pair as described above (Fig. 4b).

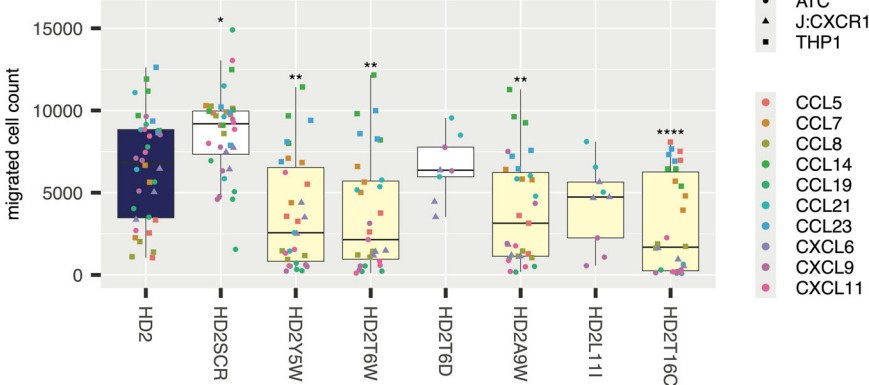

**Fig. 3 | Effect of HD2 mutant peptides on chemokine induced cell migration.**
Box-whisker plot showing impact of the indicated HD2 mutation (X-axis) upon migrated cell count (Y-axis) for all chemokines studied. Each plot shows the median as centre, 25th and 75th percentile as bounds, and 1.5*interquartile range as whiskers. HD2SCR is a scrambled version of HD2. All experiments were performed as three biological replicates and represent a meta-analysis of individual experiments provided in Supplementary Figs. 1 to 3. Individual biological replicate data points are indicated and coloured by the chemokine used. Statistically significant differences (compared to control), using a two-sided Dunnett's test with correction for multiple comparisons, are indicated by asterisks: * $P \leq 0.05$, ** $P \leq 0.01$, **** $P \leq 0.0001$. Number of biologically independent experiments per group and exact $P$-values are provided in Supplementary Table 9b. The control box in each panel is coloured blue, while boxes showing a negative value for difference from control (identified from Dunnett's test) are shown as yellow.

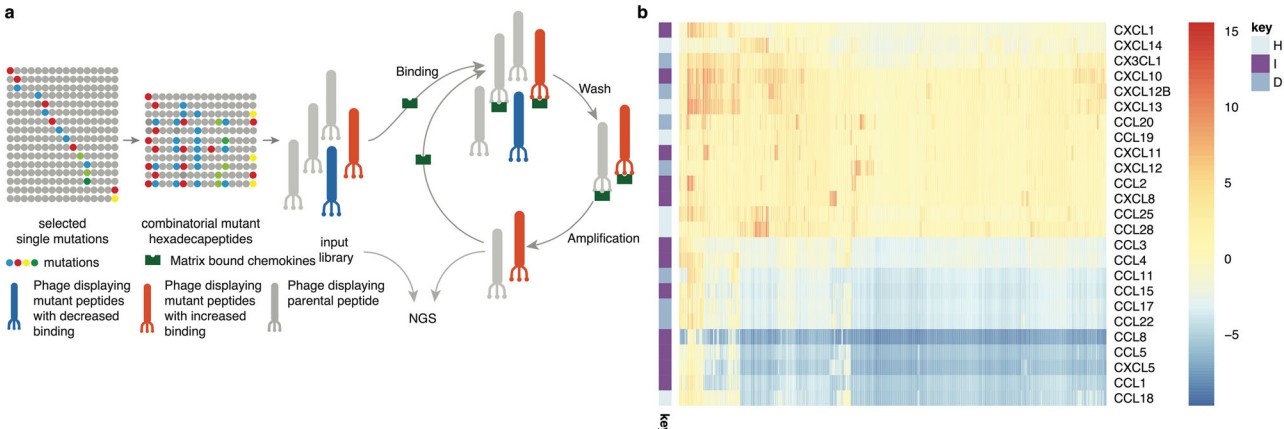

**Fig. 4 | Phage display-combinatorial mutagenesis. a** All possible combinations of 16 selected single mutations (coloured using the Taylor scale[93]) at 11 residue locations were combined to create a phage-display library with 3585 mutation combinations. The library included the parental HD2 sequence and single mutations as well. Library cloning, screening and analysis was performed as described in Fig. 1. **b** Clustered heatmap showing impact of all single and combinatorial (CM) mutations on Δlog2E compared to parental HD2 for all chemokines studied. The single mutations and parental HD2 were included in the combinatorial library, allowing direct comparisons to be made. Chemokine types in rows are indicated as I = inflammatory, D = dual, H = homeostatic. Colour scale bar indicates Δlog2E compared to parental peptide HD2.

## Identification of combinatorial mutants affecting chemokine binding

We calculated the median Δlog2E for each mutant over all chemokines to rank the data. We used the median rather than the combination of mean and peak Δlog2E used for ranking point mutations as the objective here was to identify chemokine binding peptides which have the maximum breadth of binding. To identify pan-chemokine binding peptides, we selected 5 combinatorial mutants with the highest median Δlog2E over all chemokines (CM-max). As a comparator, we also identified the 5 combinatorial mutants with the *lowest* median Δlog2E over all chemokines (CM-min). To explore if we could direct the binding towards CC and towards CXnC chemokines, we identified combinatorial mutants that had the highest median Δlog2E over CC-chemokines (CM-maxCC) and highest median Δlog2E over CXC/CX3C chemokines (CM-maxCXnC). The results are visualised in the tile-plot (Fig. 5a). We next aligned the worst and the best performing combinatorial mutation peptide sequences and generated sequence logos (Fig. 5b). The numbers of mutations in the best performing five peptides range from 5 substitutions with HD2CM307 to 7 substitutions with HD2CM539. This resembles the numbers of mutations in worst performing five peptides, with HD2CM418 having 5 and HD2CM3085 having 7 substitutions. We were unable to identify any similarity between the worst performing sequences except at residues D3, D4, and C13-F15 which had not been mutated in the combinatorial library. The worst performing sequences included several point mutations that enhanced binding in the NNK-mutagenesis phage-display experiment. Examples include Y5W in HD2CM1598 and HD2CM3085, T6W in HD2CM418 and HD2CM1516, L11I in HD2CM2132, A7D in HD2CM1598 and HD2CM2132, and T16C in HD2CM1516, HD2CM1598 and HD2CM2132. This indicates that certain mutation combinations act in an antagonistic manner. In contrast the best performing five peptides were very similar to one another, with uniform replacements at Y5W, T6D, and L11I and variable replacements at E1D, E2D, A7D, and T16D. This indicates that these mutations can act either additively or co-operatively, despite some of them (E1D, E2D, T16D) only having small and non-significant enhancement of binding in the NNK-mutagenesis phage-display experiment (Fig. 2).

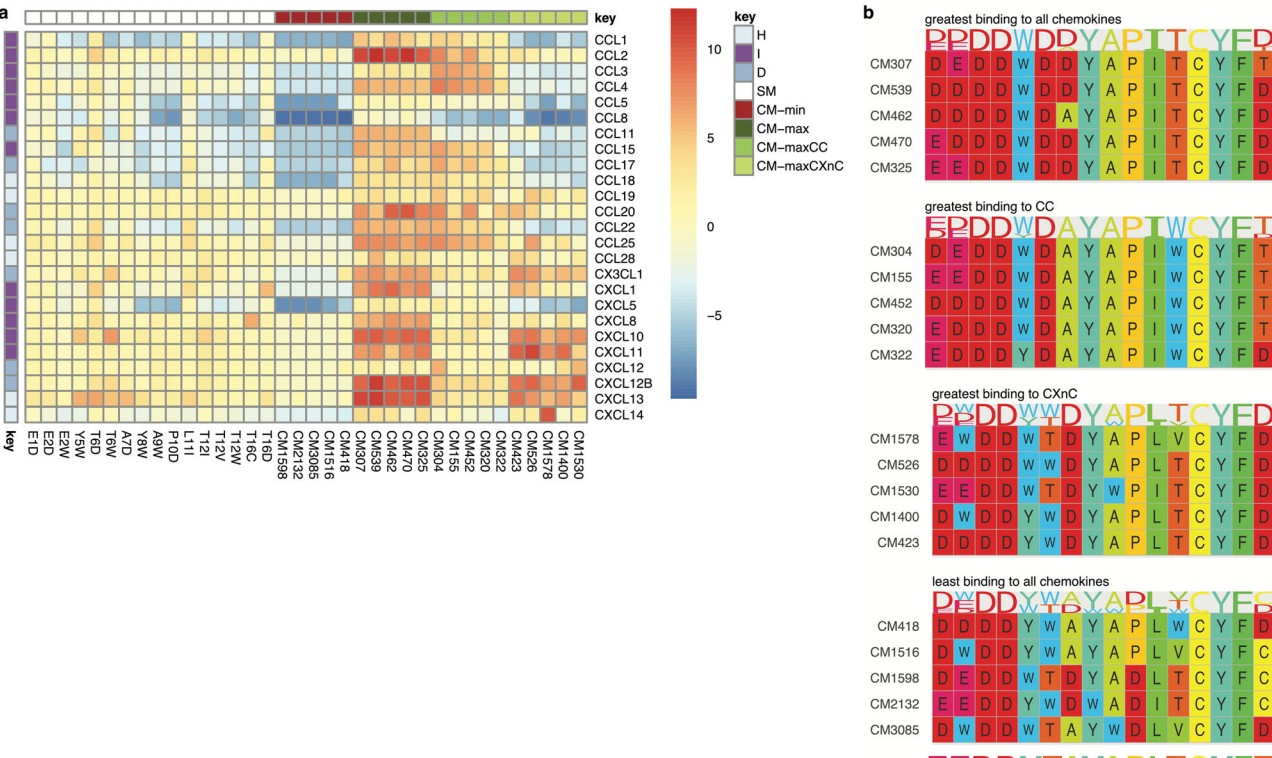

**Fig. 5 | Phage display analysis of combinatorial HD2 mutagenesis library. a** Tile-plot showing impact of 16 single and 20 selected combinatorial (CM) mutations on Δlog2E compared to parental HD2 for all chemokines studied. The 16 single mutations and parental HD2 were included in the combinatorial library, allowing direct comparisons to be made. Chemokine types in rows are indicated as I = inflammatory, D = dual, H = homeostatic. Peptide types in columns are indicated as SM = single mutants, CM = combinatorial mutants, CM-min = CMs with least binding to all chemokines, CM-max = CMs with greatest binding to all chemokines, CM-maxCC = CMs with greatest binding to CC chemokines, CM-maxCXnC = CMs with greatest binding to CXnC chemokines. Colour scale bar indicates Δlog2E compared to parental peptide HD2. **b** Sequence alignments and derived logos for the indicated CM peptides. Residues are coloured using the Taylor scale[93], and the parental HD2 sequence placed below the alignments for comparison.

## Identification of combinatorial mutants selectively affecting chemokine binding

As CC and CXnC chemokines have distinctive quaternary structures[7], we next examined the data to identify mutation combinations that are relatively selective for CC-chemokine and for CXnC-chemokine binding. We identified the five best combinations in each category (Fig. 5). The distinction in CC versus CXnC binding is apparent in the tile-plot (Fig. 5a). These results suggest that the combinatorial mutagenesis strategy can potentially identify mutation combinations that directionally enhance CC- or CXnC-chemokine binding. We next aligned the CC and CXnC-selective combinatorial mutation peptide sequences (Fig. 5b). The CC-binding sequences are characterised by constant substitutions T6D, L11I and T12W, and variable substitutions E1D, E2D, and Y5W. The CXnC-binding sequences are characterised by constant substitutions A7D, L11I and T16D, and the variable substitutions E1D, E2D, E2W, Y5W, T6W, and T12V. Thus, it appears feasible to identify combinatorial mutations that potentially distinguish CC and CXnC-class chemokines.

## Functional analysis of combinatorial mutations

We next sought to explore the effect of combinatorial mutations on biological activity. To achieve a breadth of phage-display ranking strategies in these experiments, we selected HD2CM307 and HD2CM539 from the CM-max group, HD2CM304 and HD2CM452 from the CM-maxCC group, and HD2CM526 from the CM-maxCXnC group. We studied these combinatorially-mutated peptides (HD2CM-peptide) and the point mutant peptides, using chemotaxis assays with a range of CC and CXC chemokines including CCL2, CCL3, CCL7, CCL8, CXCL10, CXCL11, and CXCL12

(Fig. 6, Supplementary Figs. 4 and 5). In each case we compared the results to the mutant HD2 peptide HD2Y5W, as this mutation was present in each of the combinatorial mutant peptides analysed and had improved inhibitory potency over parental HD2 over a range of chemokines (see above, Fig. 3). In several cases we observed statistically significant improvement of HD2CM-peptide activity in inhibiting chemokines (Supplementary Figs. 4 and 5). We next examined the inhibitory potency of HD2CM3085, a combinatorially-mutated peptide from the CM-min group (Supplementary Fig. 4e and f). This peptide contains the significantly "improving" point mutations (Y5W, A7D) and other "improving" point mutations (E1D, E2W, A9W, P10D, T12V, and T12D). However, it performs either no differently, or significantly worse than the parental peptide for inhibiting CCL7 and CCL8 migration respectively, whereas CM307 performs better than the parental peptide (Supplementary Fig. 4e and f). To achieve an overview of improvements across all CC and CXnC chemokines tested, we performed a meta-analysis of these results (Fig. 6). This showed that CM307 significantly improved chemokine inhibition across a range of CC and CXnC chemokines tested when compared to the point mutant Y5W. Taken together, the functional studies support the notion that our two-step mutagenesis and phage-display ranking strategy using median Δlog2E is an efficient method for selecting combinatorially mutated peptides that have enhanced biological efficacy.

## Potential for therapeutic application

To assess the potential of combinatorially mutated peptides as therapeutic agents in inflammatory disease, we explored their efficacy against pools of chemokines known to be produced in diseased tissues. We developed the

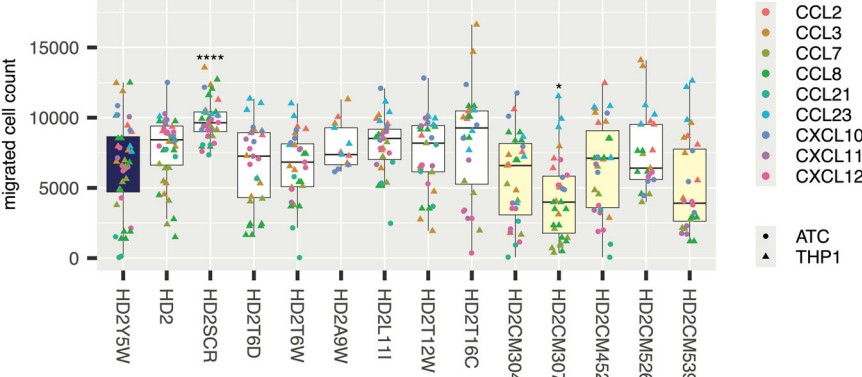

**Fig. 6 | Effect of HD2 mutations on chemokine induced cell migration.** Box-whisker plot showing impact of HD2 mutation (X-axis) upon migrated cell count (Y-axis) for the indicated chemokines. Each plot shows the median as centre, 25th and 75th percentile as bounds, and 1.5*interquartile range as whiskers. All experiments were performed as three biological replicates and represent a pooled meta-analysis of individual experiments provided in Supplementary Figs. 4 and 5. Individual biological replicate data points are indicated and coloured by the chemokine. Statistically significant differences (compared to control), using a two-sided Dunnett's test with correction for multiple comparisons, are indicated by asterisks: * $P \leq 0.05$, **** $P \leq 0.0001$. Number of biologically independent experiments per group and exact $P$-values are provided in Supplementary Table 9c. The control box in each panel is coloured blue, while boxes showing a negative value for difference from control (identified from Dunnett's test) are shown as yellow. *Abbreviations:* ATC - activated T-cells, THP1 – THP1 cells. HD2SCR indicates scrambled HD2 peptide.

chemokine pool approach to represent the naturally occurring mix of chemokines present in diseased tissues as chemokines have well-known synergistic interactions that cannot be recapitulated when studied individually[8,11,55–60]. We chose three diseases, atherosclerotic plaque, rheumatoid arthritis, and type 1 diabetes as exemplars, as there is considerable evidence supporting the role of chemokines in their pathology[2–4]. To develop disease-relevant chemokine pools, we identified well-characterised bulk RNA-sequencing datasets from atherosclerotic plaque (GSE198600) and rheumatoid arthritis synovium (GSE198520)[61,62]. As a bulk RNA-sequencing dataset for type 1 diabetes pancreatic islets could not be identified, we used, as proxy, a bulk RNA-sequencing dataset of primary pancreatic donor islets stimulated ex vivo with cytokines (GSE226888, referred to as "insulitis"), as this is thought to reflect the mechanism of islet inflammation and chemokine production in vivo[63–68]. We analysed chemokine expression levels (in transcripts per million) from these datasets (Supplementary Fig. 6). For each disease we designed a synthetic chemokine pool such that individual chemokine molar concentrations were at ratios determined by the mean relative expression of chemokines in the RNA-sequencing dataset. We then explored the efficacy of the parental peptide HD2, its scrambled control, selected individual point mutants, and the combinatorial mutants in inhibiting chemotaxis of primary leucocytes (granulocytes, monocytes and lymphocytes from donor buffy coats) and activated T-cells (ATC) by the atherosclerosis, insulitis, and rheumatoid arthritis pools (Supplementary Figs. 7-9 respectively). In these experiments we compared the output to the scrambled control as the aim here was not to demonstrate an improvement over the parental peptide HD2, but rather to assess if we had achieved, using combinatorial saturation mutagenesis, a peptide that would be effective against a chemokine pool. In these experiments we also assessed the effect of EVA4, the parental CC-chemokine binding evasin from which the HD2 series was derived[37]. EVA4 was used at equimolar concentration to the peptides. For the atherosclerosis pool, we observed near complete inhibition of ATC migration with HD2CM307 (Supplementary Fig. 7). We also observed significant inhibition of lymphocyte migration with HD2CM304 and HD2CM452 (Supplementary Fig. 7). For the insulitis pool, we observed significant inhibition of ATC migration by all CM-peptides and monocyte migration by HD2CM307 and HD2CM539. In the case of HD2CM304, ATC-migration was inhibited to near baseline levels, and substantially exceeded inhibition achieved by the parental evasin EVA4 (Supplementary Fig. 8). For the rheumatoid arthritis pool, we observed significant inhibition of lymphocyte migration by

HD2CM304 and HD2CM452 (Supplementary Fig. 9). To have an overview of improvements in the different pools across different cell types, we performed a meta-analysis of these results (Fig. 7). For the atherosclerosis and insulitis pools, significant inhibition of chemotaxis was observed across most CM-peptides, with HD2CM304 and HD2CM307 being most potent in inhibiting chemotaxis (Fig. 7a, b). For the rheumatoid arthritis pool, HD2CM304 had the greatest effect (Fig. 7c).

We next determined the potency of HD2CM304 and HD2CM307 and compared them to the potency of the parental HD2 peptide. We performed dose-response experiments for each inhibitor against each chemokine pool. We performed these experiments using either activated T cells (ATC), which migrate strongly in response to CXC chemokines, or THP1 cells, which migrate strongly in response to CC chemokines (see Supplementary Table 1) to determine $IC_{50}$. Example peptide dose-response curves for each chemokine pool and cell type are shown in Supplementary Figs. 10 and 11. For summary statistics we converted $IC_{50}$ to $pIC_{50}$ (negative $\log_{10}$ of $IC_{50}$ expressed in moles/litre ([M]) and displayed them as box-whisker plots (Fig. 8) with summary statistics provided in Supplementary Table 2. These results show that HD2CM304 had a significantly higher $pIC_{50}$ in comparison to HD2 when tested against three different chemokine pools using ATC as the responding cell type. For certain chemokine pools (e.g., RA) the improvement in $pIC_{50}$ is by over an order of magnitude, with mean $pIC_{50}$ improving from 5.02 to 6.35 [M], corresponding to an $IC_{50}$ change from 9.5E-6 to 4.5E-7 [M]. There was no significant difference between $pIC_{50}$ for HD2 and HD2CM304 when tested against THP1 cells. HD2CM307 showed improvement of $pIC_{50}$ over HD2 for all three chemokine pools when tested using THP1 cells but this was statistically significant only for the ATHERO pool. It showed no significant difference from parental HD2 when tested using activated T-cells. The intriguing pool and cell-specific differences observed likely result from variation in synergistic chemokine interactions in different pools, and the ability of different cell-types to respond to chemokines within the pool.

**Structure-function analysis**

We next examined potential structural mechanisms that may underlie the enhancement of potency. Our previous studies using AlphaFold2-Multimer[69] had indicated that HD2 occludes distinct receptor-binding regions in CC and CXC-chemokines, and that the docking predictions are consistent with NMR-based models[37]. We generated models (Fig. 9a) of 46 chemokines in complex with HD2, HD2CM304 or HD2CM307 using

**Fig. 7 | Effect of HD2 mutant peptides on chemokine pool-induced cell migration.** Box-whisker plots showing impact of HD2 mutant peptides (X-axis) upon migrated cell count (Y-axis) for disease chemokine pools from **a** atherosclerotic plaque ("ATHERO"), **b** cytokine-stimulated islets ("INS"), and **c** rheumatoid arthritis synovium ("RA"). Each plot shows the median as centre, 25th and 75th percentile as bounds, and 1.5*interquartile range as whiskers. All experiments were performed as three biological replicates. Individual biological replicate data points are indicated and coloured by the cell type. Statistically significant differences (compared to control), using a two-sided Dunnett's test with correction for multiple comparisons, are indicated by asterisks: * $P \le 0.05$, ** $P \le 0.01$, ***, $P \le 0.001$, **** $P \le 0.0001$. Number of biologically independent experiments per group and exact $P$-values are provided in Supplementary Table 9d. The control box in each panel is coloured blue, while boxes showing a negative value for difference from control (identified from Dunnett's test) are shown as yellow. *Abbreviations*: ATC - activated T-cells, GR - granulocytes, LY - lymphocytes, MO - monocytes. HD2SCR indicates scrambled HD2 peptide.

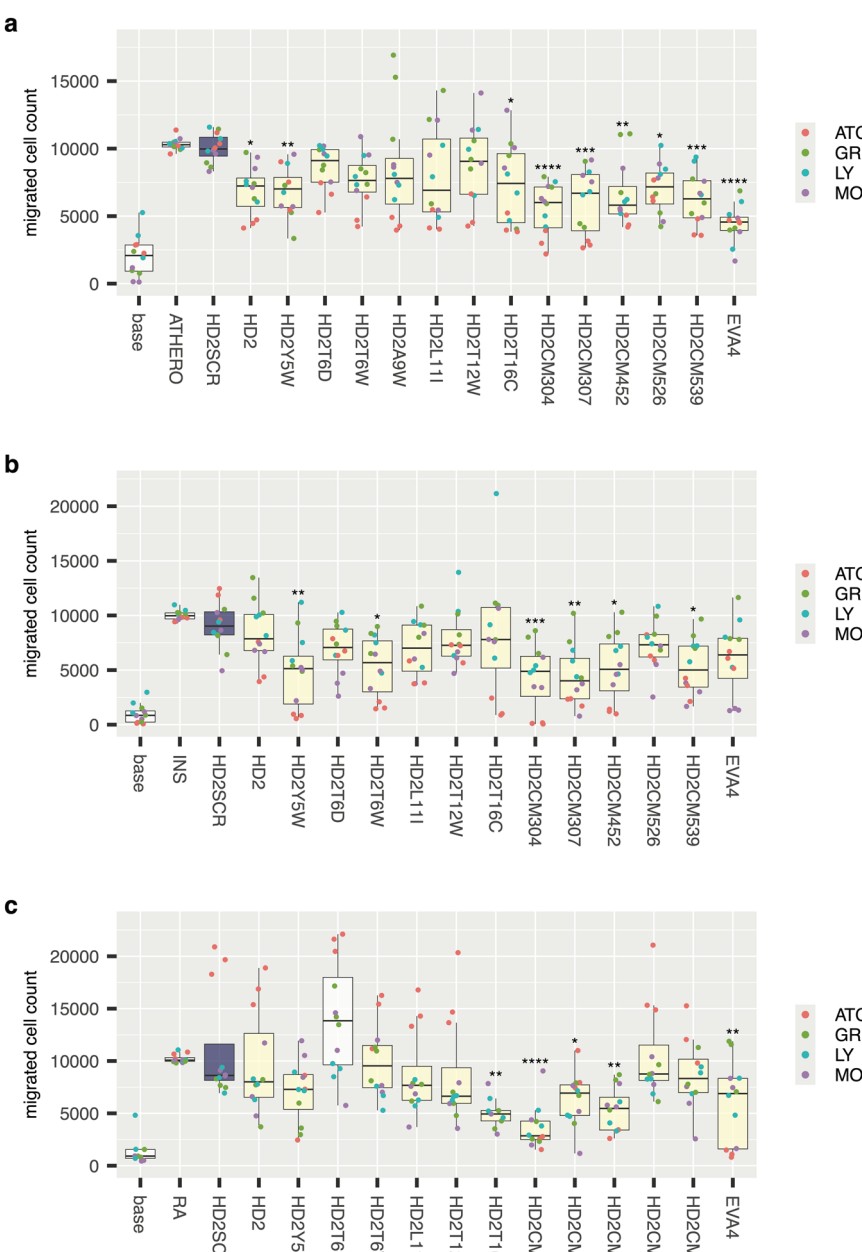

AlphaFold 3, which predicts the joint structure of complexes and performs better than AlphaFold2-Multimer[70]. These models are consistent with previously generated models of HD2 binding to CC and CXC chemokines[37]. Using the AlphaFold 3 models, we identified interchain atomic bonds between the peptide to the chemokine using Arpeggio[71], and generated summary statistics by peptide. These results indicate that the total number of bonds formed by HD2CM304 and HD2CM307 significantly exceed the total number of bonds formed by HD2 (Fig. 9b, subpanel "all"). We also examined each bond class i.e. aromatic, carbonyl, hydrophobic, hydrogen, ionic, polar or van der Waals by peptide (Fig. 9b, individual subpanels). This analysis indicates that not only do hydrophobic bonds constitute most peptide - chemokine inter-chain bonds, but that there is a significant increase in the numbers of hydrophobic bonds in the case of HD2CM304 or HD2CM307 (Fig. 9b, subpanel "hydrophobic").

We next generated the summary statistics by residue to investigate if there were residue-level differences in hydrophobic bonding (Fig. 9c). This analysis indicated that the total numbers of hydrophobic inter-chain bonds formed at residues 5 and 12 by HD2CM304 significantly exceed those

formed by HD2 (Fig. 9c, middle panel). For HD2CM307 a significant difference was only observed at residue 5 (Fig. 9c, bottom panel). Residues 5 and 12 are both mutated to Trp (W) in HD2CM304, whereas only residue 5 is mutated to Trp in HD2CM307. These results suggest that increased hydrophobic bond formation, primarily mediated by Trp residues at positions 5 and 12, could contribute to the enhanced $pIC_{50}$ of HD2CM304 compared to HD2.

**Mechanism of chemokine inhibition**

We next examined the impact of peptide binding on chemokine interactions known to play a functional role in immune cell migration. The most important interactions are thought to occur between chemokines and their cognate receptors[11], and chemokine dimerization, i.e. with other chemokines[8]. Using AlphaFold 3, we generated 80 models of chemokines in complex with their cognate receptors. For these models, chemokine – receptor interactions with documented signaling activity (i.e. cognate receptors) were identified from the IUPHAR database[72]. We also generated 64 models of chemokine dimers using AlphaFold 3. Chemokine dimer

**Fig. 8 | Effect of HD2 and combinatorial mutant peptides on pIC50.** Box-whisker plots showing impact of HD2 mutant peptides (X-axis) upon pIC$_{50}$ (negative log$_{10}$ of IC50 expressed in [M], Y-axis) for disease chemokine pools from **a** atherosclerotic plaque ("ATHERO"), **b** cytokine-stimulated islets ("INS"), and **c** rheumatoid arthritis synovium ("RA"). Each plot shows the median as centre, 25th and 75th percentile as bounds, and 1.5*interquartile range as whiskers. All experiments were performed as three biological replicates. Plots display median, lower and upper hinges (25th and 75th percentiles), and whiskers from hinge to 1.5* interquartile range. Left panels: Activated T-cells (ATC). Right panels: THP1 cells. Individual biological replicate data points are indicated. Statistically significant differences (compared to control), using a two-sided Dunnett's test with correction for multiple comparisons, are indicated by asterisks: * $P \leq 0.05$, ** $P \leq 0.01$. Number of biologically independent experiments per group and exact $P$-values are provided in Supplementary Table 9e. The control box in each panel is coloured blue.

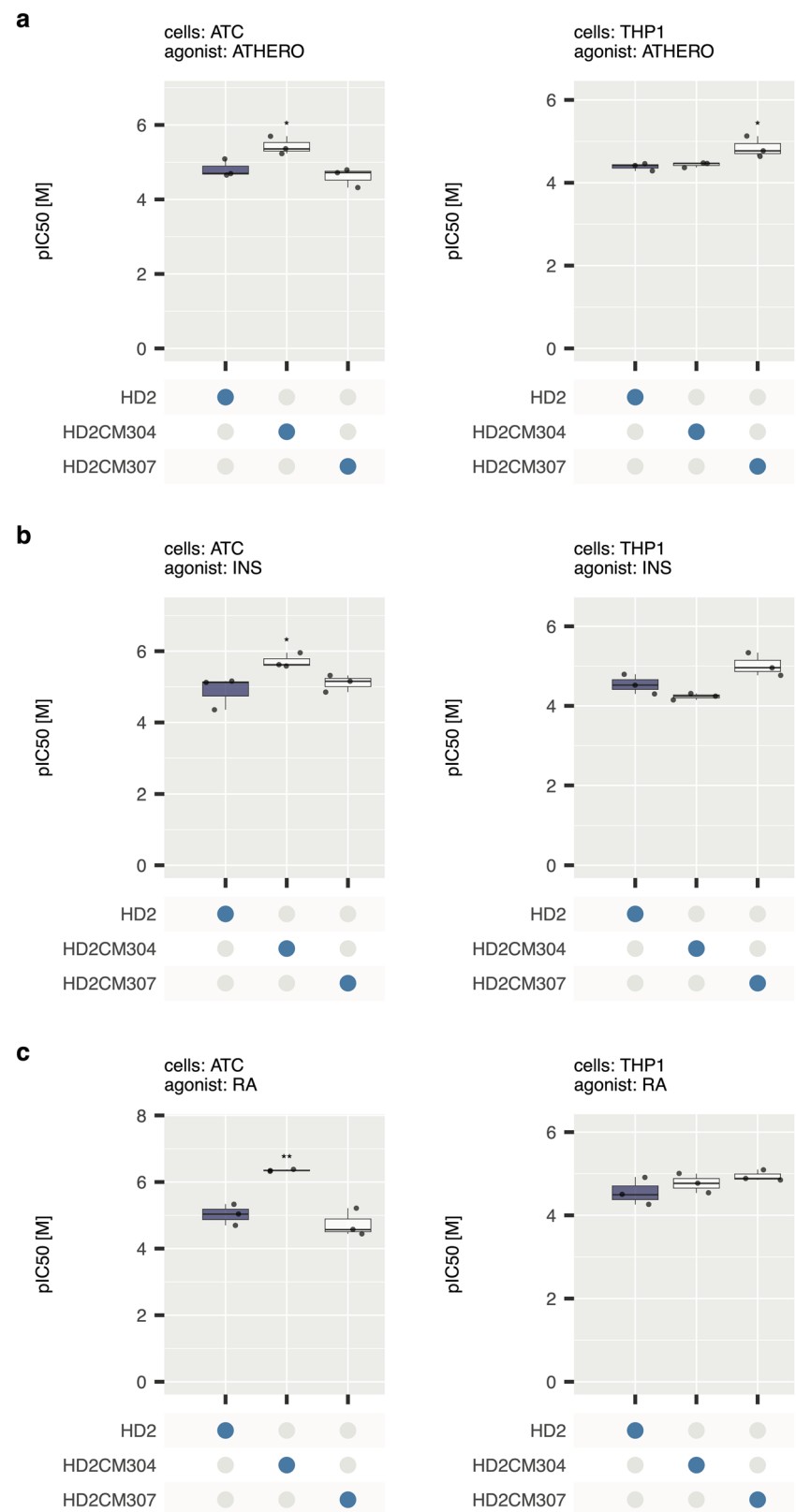

partners were identified from the PSICQUIC database[73]. We restricted our analyses to those dimers that are supported by evidence from surface plasmon resonance, x-ray crystallography or NMR experiments[73]. Examples of these models are shown in Figs. 10a, b and are similar to previously reported chemokine dimer models[8]. For each chemokine, we identified, using Arpeggio[71], binding site residues, i.e. those within 5 Å of either a cognate receptor or a chemokine dimer partner. We then determined, for each chemokine, the proportion of these residues that were predicted to be occupied (i.e. within 5 Å) by either HD2 or HD2CM304. These analyses are shown as stacked bar plots in Figs. 10c, d respectively. The plots indicate that

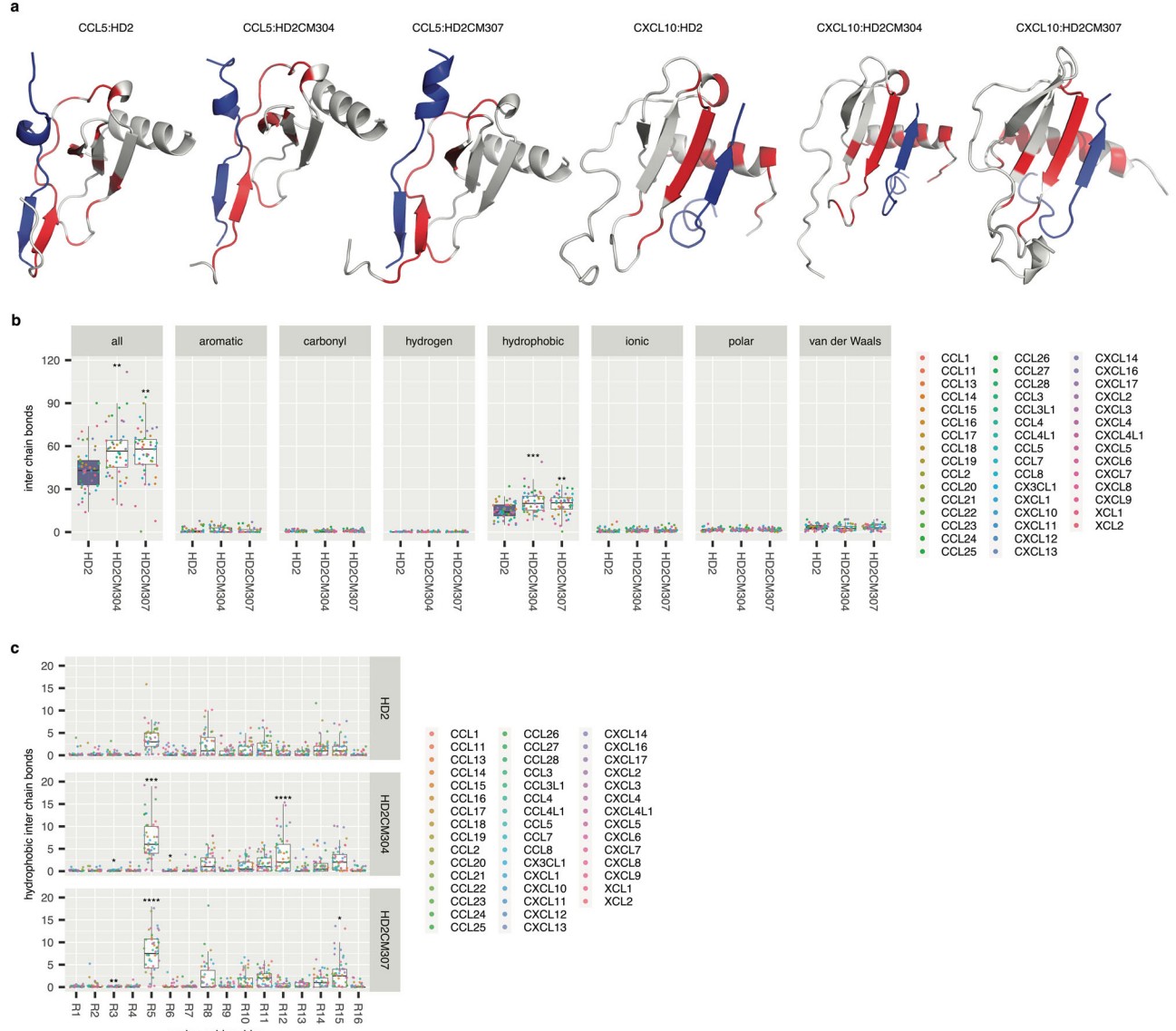

**Fig. 9 | Effect of HD2CM304 and HD2CM307 mutations on peptide - chemokine bonds. a** Exemplar models of 138 chemokine:peptide complexes generated using AlphaFold 3. Chemokines are coloured grey and the peptide in blue. The peptide binding site on the chemokine is coloured red. **b** Faceted box-whisker plots showing numbers of inter-chain bonds (Y-axis) for HD2, HD2CM304 and HD2CM307 peptides (X-axis) with indicated chemokines. Each plot shows the median as centre, 25th and 75th percentile as bounds, and 1.5*interquartile range as whiskers. Individual facets show data for all bonds ("all") or by bond type. Individual data points are coloured by chemokine. Control boxes in each panel are coloured blue. **c** Faceted box-whisker plot showing numbers of inter-chain hydrophobic bonds (Y-axis) for HD2, HD2CM304 and HD2CM307 peptides, displayed by residue (X-axis). Statistically significant differences (compared to control HD2), assessed using a two-sided Dunnett's test with correction for multiple comparisons, is indicated by asterisks: * $P \leq 0.05$, ** $P \leq 0.01$, ***, $P \leq 0.001$, **** $P \leq 0.0001$. Number of biologically independent observations per group and exact $P$-values are provided in Supplementary Table 9f.

a substantial number of chemokine interactions, involving chemokine-receptor binding and chemokine dimerization, are blocked by HD2 and by HD2CM304. Taken together these results suggest that HD2 and HD2CM304 function to inhibit immune cell migration by disrupting chemokine function.

## Discussion

In this study we have explored the use of multiplexed phage-display selection to identify peptide mutations and combinations of mutations that increase affinity to multiple structurally related targets. Our approach used saturation mutagenesis of the lead peptide HD2 to identify improving point mutations as an initial step, followed by a second step where all possible combinations of point mutations were screened. We show that this two-step method allows efficient selection of combinatorially mutated peptides that enhance binding across a range of chemokines in phage-display. We also

show that the selected combinatorially mutated peptides have a significant improvement in chemokine-inhibiting potency, not only when studied against individual chemokines, but also when evaluated against a pool of disease-expressed chemokines.

Our studies of the point mutants indicated that even single residue changes e.g. Y5W, T6W, A9W and T16C can enhance chemokine binding and chemokine-inhibiting potency. The mutation Y5W led to de novo inhibition of chemokines e.g. CCL19, CCL21 and CXCL11 that are not inhibited by the parental HD2. Tryptophan (W) is frequently found in protein-protein interaction "hot-spots" and creates hydrophobic interactions, π-interactions, and hydrogen bonding with multiple residues[74,75]. The most likely explanation of our results is the improvement of chemokine-binding afforded by the substitution with tryptophan at a critical hotspot region of HD2. The Y5 "hot-spot" is within a conserved sequence motif of HD2, and mutation to alanine significantly reduced binding in phage

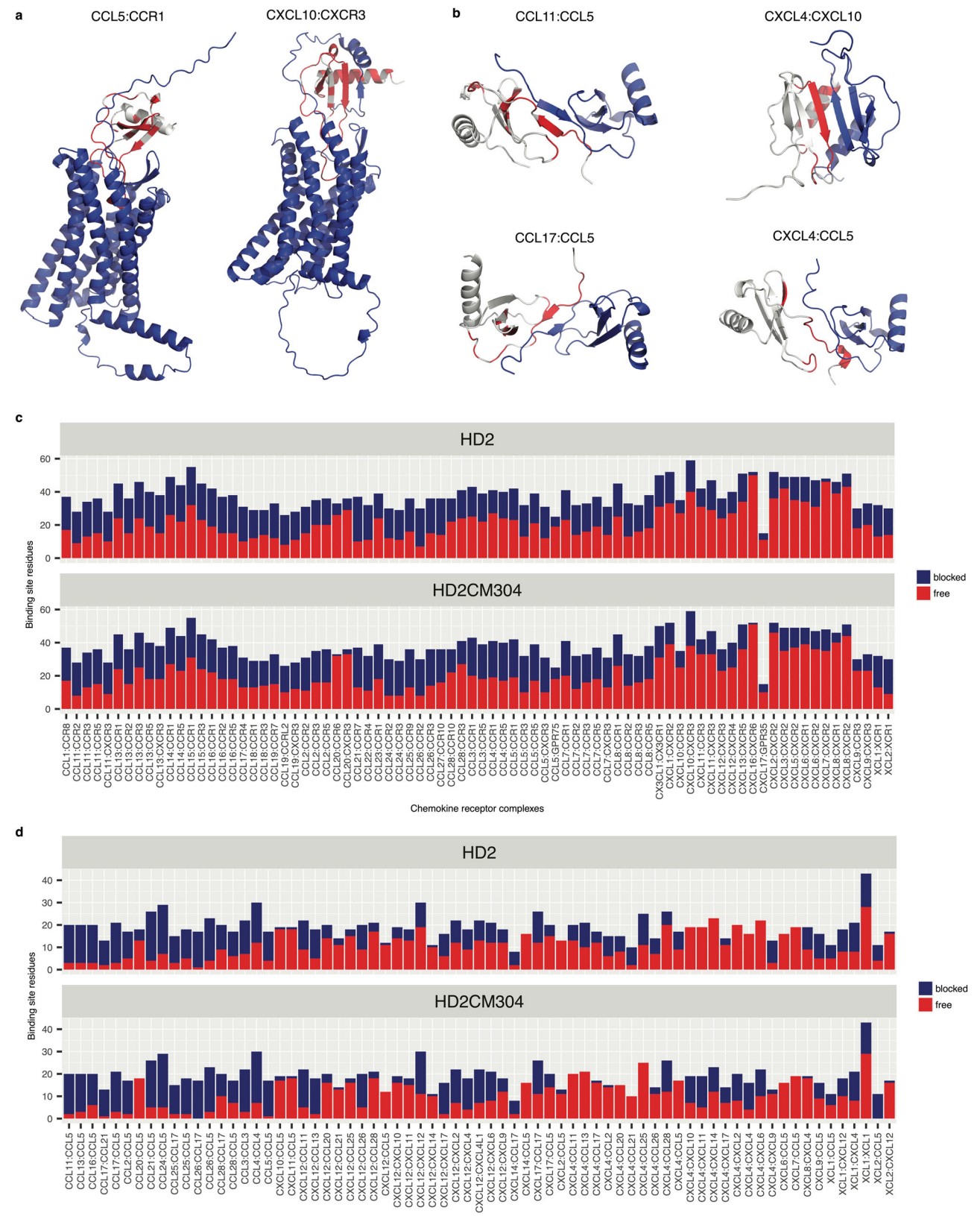

**Fig. 10 | Occlusion of functional chemokine residues by peptides. a** Examples of 80 AlphaFold 3 models of chemokines complexed with cognate receptors. Chemokines are coloured grey and the receptor in blue. The receptor binding site on the chemokine is indicated in red. **b** Examples of 71 AlphaFold 3 models of chemokine dimers. Chemokines are coloured grey and the dimer partner blue. The dimer partner binding site on the chemokine is indicated in red. **c** Stacked bar charts showing total number of binding site residues (Y-axis) for each chemokine:receptor pair (X-axis). For each chemokine, binding site residues were defined as those within 5 Å of the partner receptor. The residues were classified as "blocked" if they were also within 5 Å of the indicated peptide in the corresponding model. **d** Stacked bar charts showing total number of binding site residues (Y-axis) for each chemokine:chemokine dimer pair (X-axis). For each chemokine, binding site residues were defined as those within 5 Å of the partner chemokine. The residues were classified as "blocked" if they were also within 5 Å of the indicated peptide in the corresponding model.

display[37]. Phage-display also allowed us to identify functionally "improving" mutations e.g. T6W, at a site previously shown not to influence binding or potency by alanine-scanning mutagenesis[37]. While significant gains in potency were observed with the point mutants for several chemokines, there was no significant loss of potency for any chemokine, indicating that peptide specificity was being broadened rather than shifted.

Analysis of the combinatorial mutants by phage-display indicated that it is possible to select mutation combinations that have enhanced chemokine binding (e.g. HD2CM307) and identify others (e.g. HD2CM3085) that do not. The mutation T16C was not present in the best five peptides but was frequently present in the worst five peptides suggesting that it has antagonistic interactions with other single mutations, at least in the context of phage-display. These results indicate that certain individually "improving" point mutations have co-operative or additive effects in combination, whereas others have antagonistic effects. The molecular mechanisms underlying such effects in the case of peptides identified in our study are unclear. Given the multiple mutations introduced in several peptides, their elucidation will likely require structural approaches.

Importantly, the phage-display selection method positively correlated with the ability to inhibit chemotaxis. Studies using cell migration assays indicated that the combinatorial mutants (HD2CM307 and HD2CM304) selected for improved chemokine binding, had an inhibitory effect not only when tested against individual chemokines, but also when tested against pools of chemokines known to be expressed in disease. We used a chemokine pool approach to allow the modelling of the naturally occurring mix of chemokines present in diseased tissues to recapitulate well-known synergistic chemokine interactions[8,11,55–60]. To better model the in vivo diseased tissue environment we used primary white blood cells and activated T-cells rather than cell lines. In certain cases, the combinatorially mutated peptides outperform the parental evasin EVA4, a protein known to be effective in several models of inflammatory disease, at equimolar doses[28,29,33]. In these chemokine pool experiments we used bulk RNA data as an unbiased proxy for the relative amounts of chemokine protein produced by all cells in the diseased tissue. We chose not to use single-cell RNA sequencing data to guide the design of the chemokine pool as this does not provide information regarding total chemokine production by all cells in a tissue. The RNA-based approach has limitations. While transcript levels generally correlate with protein, there is likely to be substantial variation in protein expression due to post-transcriptional mechanisms[76,77]. A preferred approach in the future may be to use unbiased proteomic approaches to obtain quantitative measurements of chemokine proteins in diseased tissues and use these to model the in vitro chemokine pool. In vitro assays of course cannot entirely replicate the full complexity of the mammalian immune system, and the further development of the combinatorially mutated peptides into therapeutics described here will require proof-of-concept experiments in suitable animal models of inflammatory disease.

To confirm these findings, we performed dose-response studies, comparing HD2, HD2CM304 and HD2CM307 to determine the pIC50, against chemokine pools from three different diseases. In these experiments we used activated T-cells which migrate strongly in response to CXC chemokines, or the THP1 cell line, which migrates strongly in response to CC chemokines. These experiments revealed that while HD2CM304 was consistently significantly superior to the parental HD2 when tested against activated T-cells, there was no significant difference from the parental HD2 when tested against THP1 cells. HD2CM307 showed significantly improved activity over HD2 when tested using THP1 cells and the ATHERO pool. These results underscore the importance of performing experiments against different pools of chemokines to model disease pathophysiology in vitro.

As both HD2CM304 and HD2CM307 showed significant improvement over parental HD2 in the dose-response experiments, we next sought to understand the molecular mechanism. We performed a structure-function analysis using a state-of-art approach – AlphaFold 3 – and generated models of the three peptides in complex with 46 chemokines. We analysed the models using Arpeggio[71] to identify interactions at the chemokine-peptide interfaces. These results showed that the number of inter-chain bonds, and particularly hydrophobic bonds formed by HD2CM304 and HD2CM307 far exceeded the numbers formed by the parental HD2. As hydrophobic interactions are increased in high-efficiency ligands[78] this suggests that the increased numbers of hydrophobic interactions, at least in part, explains their greater effectiveness over HD2 in cell-migration experiments. Our results also suggest that the distribution of hydrophobic interactions across the peptide could be important in explaining the differences between HD2CM304 and HD2CM307 in the cell-migration experiments. While both peptides have similar numbers of hydrophobic inter-chain bonds, in HD2CM304 they arise from the Trp-substituted residues 5 and 12, replacing Tyr and Thr respectively, whereas in HD2CM307 they arise from Trp-substitution of Tyr at residue 5. Trp is classed as a large volume hydrophobic residue, whereas Tyr and Thr are neutral with regard to hydropathy[79], explaining how Trp substitutions at these sites result in increased hydrophobic bond formation.

We next investigated if binding of either HD2 or HD2CM304 to chemokines would sterically hinder chemokine interaction with cognate chemokine receptors or with chemokine dimerization. We found that a substantial number of chemokine residues that form protein interaction surfaces, both with cognate receptors and with dimeric partners, are predicted to be occluded following HD2 or HD2CM304 binding. Thus, it is likely that these peptides, at least in part, function by sterically hindering chemokine interactions that are required for immune cell migration.

Taken together, these results indicate that our two-step CoSMOS method is an efficient way of identifying the best "improving" mutations and combinations thereof without recourse to the structural characterisation of peptide interactions with multiple network members. The significant advance of our approach is that it does not depend on an intermediate functional validation step of point mutant peptides and can substantially shorten the therapeutic peptide development process. The broad-spectrum chemokine-network inhibiting peptides described here could have applications in the therapy of a range of immune-inflammatory diseases where chemokines play a critical role, including atherosclerosis, type 1 diabetes, rheumatoid arthritis and inflammatory bowel disease[2–5]. We envisage that future developments would take advantage of molecular "ZIP-codes" to target such agents to the disease site, minimising undesirable chemokine pathway neutralisation elsewhere[80]. Our approach could be broadly applicable for the rapid identification of combinatorially mutated peptides that target other disease networks characterised by multiple structurally related members and redundant signalling pathways. These include immune-inflammatory disease networks driven by structurally related cytokines such as the TNF, IL1, IL6 and IL17 families[81–84].

## Methods

### Human blood samples
We obtained peripheral blood cells from anonymized donor leucocyte cones and buffy coats purchased from NHS Blood Transfusion Services. As donor samples were anonymized at source, gender, race, ethnicity, population, social characteristics and recruitment data were not available. Ethics approval was obtained from the University of Oxford Medical Sciences Interdivisional Research Ethics Committee, CUREC1 approval reference R75963/RE001. All ethical regulations relevant to human research participants were followed.

### Chemokines
Sources of biotinylated chemokines for phage-display experiments, and non-biotinylated chemokines for cell migration experiments are provided in Supplementary Tables 3-7.

### Saturation mutagenesis phage display library design and construction
81-mer mutant oligonucleotides replacing NNK at each of the 16 peptide encoding residues were designed with the sequence arms 5'-GCAGC CTCTTCATCTGGC, and GGTGGAGGATCCGGA-3' at respective ends to enable amplification and cloning. Oligonucleotides (Genscript)

were individually amplified and cloned into plasmid prSTOP4Nsi (described in[37]). prSTOP4Nsi was amplified using primers (Sigma):

pRSTOP4Nsi_fwd: 5'-GGAGGCGCCGAGGGTGAC and
pRSTOP4Nsi_rev: 5'-ATAGGCATTTGTAGCAATAGAAAAAAC GAACATAGATGCAAG.

Oligonucleotides were amplified using primers (Sigma): oligo_fwd: 5'-CTATTGCTACAAATGCCTATGCAGCCTCTTCATCTGGC and oligo_rev2: 5'-TCGTCACCCTCGGCGCCTCCTCCGGATCCTCCACC. The PCR products were cloned into prSTOP4Nsi plasmid using NEB Builder HiFi Assembly (E5520, NEB) following instructions of the manufacturer. The vector pool was used to transform electrocompetent Electro-MAX™ DH5α-E™ cells (11319019, Invitrogen). Plasmid DNA was harvested by MaxiPrep (GeneJet Plasmid Maxiprep Kit (K0491, Thermo Scientific)). DNA was transformed into SS320 phage display electrocompetent cells (60512, Lucigen), following the manufacturer's protocol. Following electroporation, cells were resuspended in 950 μL recovery media and transferred to 50 mL Falcon tubes. 100 μL 10[11] CFU/mL M13KO7 helper phage (N0315S, NEB) was added, and tubes shaken at 37 °C, 220 RPM, 1 h. 1 mL recovery culture was added to 2YT medium containing +50ug/mL carbenicillin (C9231, Sigma) and 25ug/mL kanamycin (K1377, Sigma), and grown overnight at 37 °C, 220 RPM. Phage was precipitated from culture supernatant by adding 22.5 mL of supernatant to 4.5 mL 20% PEG 8000 (89510, Sigma)/2.5 M NaCl (S9625, Sigma), incubating on ice for 30 min, and centrifuging at 12857 g, 4 °C, 10 min. The phage pellet was resuspended in 1 mL of PBT (PBS (P4417, Sigma), BSA (A7030, Sigma) 0.20%, Tween 20 (P1379, Sigma) 0.05%). Undissolved particulate matter was removed by centrifugation at 15871 g, 4 °C, 10 min. Glycerol (G5516, Sigma, 10% final (v/v) was added, and aliquots stored at −80 °C. A schematic of phage-display library construction is provided in Supplementary Fig. 12.

### Phage display library screening

Phage display screening was performed as described[37]. Experiments were performed in 96-well plates (Greiner Bio-One, 655180). Biotinylated chemokines or complement C5A (1μg, Almac or Protein Foundry) were incubated overnight at 4 °C with shaking at 250 RPM with 5μl streptavidin-coated magnetic beads (Dynabeads™ M-280 Streptavidin, Invitrogen, 11205D) in 100 μl PBS. Chemokine-bound beads were blocked for two hours in 200μl blocking buffer (PBS (phosphate-buffered saline, Sigma, P4417) + 0.2% BSA (bovine serum albumin, Sigma, A7030) with shaking at 250 RPM. Phage library (100μL, 10[10] CFU/mL) was bound to blocked beads for two hours at room temperature with shaking at 250 RPM. Beads were washed 15 times with PT buffer (PBS + 0.05% Tween 20 (Sigma, P1379)) to remove unbound phage then transferred to a fresh plate. Beads were incubated with 100μL phage-resistant Omnimax *E. coli* (Invitrogen, A10469) at optical density (OD$_{600}$) 0.6-0.8, shaken at 37 °C for 30 min. Following this 10 μL M13K07 helper phage (NEB, N0315S, final concentration 10[10] CFU/mL) was added and shaken at 37 °C for 45 min. Cells were transferred to 1 mL 2YT medium (at pH7, MP Biomedicals™, 113012031, supplemented with 150 μg/mL carbenicillin (Sigma, C9231) and 75 μg/mL kanamycin (Sigma, K1377) in a 96-deep well block with V-bottom (Corning, CLS3960) and grown with overnight shaking at 37 °C, 200 RPM. Plates were either centrifuged at 2000g (4 °C, 30 min) or transferred to screw cap tubes and pelleted by centrifugation at 4000 g and 4 °C for 15 min. 540 μL of supernatant was transferred to a new 96-deep well plate, 60 μL of 10×PBT (10×PBS + 2% BSA + 0.5% Tween 20) added and stored at 4 °C for subsequent rounds. Phage display was carried out for three rounds. A schematic of phage-display library screening is provided in Supplementary Fig. 12.

### Next generation sequencing

We amplified inserts from the input library and from the selected phage population directly from phage by PCR using primers: 5'-ACACTCTT TCCCTACACGACGCTCTTCCGATCTCTAGCGCTATGCCTATGCA GCCTCTTCA and 5'-GACTGGAGTTCAGACGTGTGCTCTTCCGAT

CTCGTCTGCGATGACAACAACCATCGCCCA (Life Technologies). For amplification we used Q5 High-Fidelity DNA polymerase (NEB, 0.02U/ μl), 200 μM dNTPs, 0.5 μM primers, 2 μL phage in a 50 μL reaction. PCR conditions were an initial denaturation step of 98 °C for 30 s, followed by 25 cycles of 98 °C/10 s, 67 °C/30 s, 72 °C/30 s, and final extension of 72 C°/ 2 min. PCR products were cleaned up using Monarch® PCR & DNA Cleanup Kit (5 μg, T1030, NEB), 260/280 ratio determined by Nanodrop, quantified using a Qubit dsDNA Quantitation (High Sensitivity Kit, Q32851, Thermo Scientific) and sequenced at Azenta/GeneWiz using the AmpliconEZ protocol.

### Next generation sequence analysis

Sequence analysis was performed as follows. The "read id" was used to join paired end read Fastq files. Only sequences containing both forward and reverse demultiplexing regions (5'-CTAGCGCT, 5'-CGCAGACG) and both constant regions (5'-ATGCCTATGCAGCCTCTTCATCTGGC, 5'-GGTGGAGGATCCGGAGGAGGCGCCGAGGGTGACGATCCCGC AAAAGCGGCCTTTAACTCCCTGCAAGCCTCAGCGACCGAATAT ATCGGTTATGCGTGGGCGATGGTTGTTGTCAT) were analysed. Insert sequences from each read were identified. Reads with non-identical or ambiguous insert sequences were excluded. Reads with incorrect insert size (i.e., not 48 nucleotides) were also excluded. Only insert peptide sequences present in the designed library were counted and frequency determined. Following selection, enrichment (E) was calculated as ratio of output to input peptide frequencies and expressed as log2E. Where proportion of count in input library was not available it was replaced with the lowest input proportion in the experiment rather than replacing it with 0. This is needed to avoid a value of E that is infinity. Where proportion of count in output was not available it was replaced with the lowest output proportion in the experiment rather than zero, which would result in a -Infinity log2E. Δlog2E was calculated as the difference between log2E of peptide and that of parental HD2. Peptides that showed binding to control (C5A, log2E > 0) were excluded from downstream analysis.

### Combinatorial mutagenesis library design

As CC and CXnC binding peptides appeared to have a different improving mutation spectrum we selected mutations using an algorithm that optimizes CC and CXnC improving mutations separately. Briefly, we identified at each peptide position the mutation giving highest mean Δlog2E and the highest peak Δlog2E. We took the following steps to limit the numbers of selected mutations. We set a threshold 5 for peak Δlog2E, and a threshold of 0.55 for mean Δlog2E, and then selected the top 10 mutations (arranged by mean Δlog2E and peak Δlog2E) for CC and top 12 mutations for CXnC chemokines. This resulted in a total of 16 mutations at 11 residues. We generated all possible combinations of the 10 CC-enhancing, and 12 CXnC-enhancing mutations using the R function "utils::combn", pooled the output, added the parental and scrambled parental sequence, and reverse translated the peptides using R package reversetranslate_1.0.0, with ecoli_tbl and model ="gc_biased" to generate 3585 oligonucleotide sequences. This included parental HD2, HD2SCR, and all point mutations. The numbers of combinatorial mutations incorporated into HD2 ranged from 2 to 11. The oligonucleotides were synthesized as a pool (Genscript, 12 K chip) and were cloned into plasmid pRSTOP4Nsi as described above. The library quality was assessed by NGS AmpliconEZ (50000 reads), and 3584 of 3585 inserts were identified. The library was screened using the phage-display protocol described above. Only insert peptide sequences present in the designed library were counted. Adjusted counts for input and output libraries were generated from raw counts by adding 1 to all counts to avoid division by zero, and frequency determined. Following selection, enrichment (E) was calculated as ratio of output to input peptide frequencies and expressed as log2E. Δlog2E was calculated as the difference between log2E of peptide and that of parental HD2. Peptides that showed binding to control (C5A, log2E > 0) were excluded from downstream analysis.

## Peptides

Lyophilised peptides were obtained from GenScript at >95% purity with a C-terminal amide. Peptide sequences expected and observed molecular weights and solubility data provided by the manufacturer are provided in Supplementary Table 8. To maintain consistency for inter-peptide comparison, all lyophilised peptides were initially dissolved in 100% DMSO at peptide concentration of 20 mM, mixed well using a vortex and then diluted in PBS to a final peptide concentration of 1 mM and 5% DMSO. With this protocol all peptides were soluble (despite certain peptides being reported as insoluble by the manufacturer) and no precipitation was observed upon addition of PBS. Upon resuspension, the peptide solutions were stored at −20 °C in aliquots until use. Peptide stoichiometry was assessed by LC-MS data provided by the manufacturer, and all peptides were monomeric as evident from observed mass (Supplementary Table 8).

## Plasmids

The EVA4 expression plasmid P1909 (EVA4_RHISA-StrepII-8xHis) was constructed in plasmid pHLSec[85] using an In-Fusion cloning kit (Takara) following the instructions of the manufacturer, and contains EVA4_RHISA mature peptide (i.e., residues 24-127) with a C-terminal StrepII- 8xHis-tag. Plasmid sequences were confirmed by Sanger sequencing (Azenta/ Genewiz UK).

## Protein Expression

EVA4 was transiently expressed in HEK293F cells (R79007, ThermoFisher Scientific) using the P1909 plasmid and polyethylenimine (24765, Polysciences). Cells were cultured in Freestyle 293 expression medium (12338026, ThermoFisher) at 37 °C, 8% CO2, and 130 rpm for five days. The protein was purified from filtered supernatants using nickel-charged IMAC Sepharose 6 Fast Flow resin. Eluted fractions were concentrated with an Amicon Ultra-15 Centrifugal Filter Unit and further purified by size exclusion chromatography (SEC) on an AKTA Start system with a HiLoad 16/600 Superdex 75 column (GE28-9893-33, GE Healthcare) in SEC buffer (PBS + 150 mM NaCl). Fractions with a 280 nm absorbance peak were analyzed via electrophoresis using a Bolt Bis-Tris Plus Mini Protein Gel, 12% (NW00122, Invitrogen), in Bolt MES SDS Buffer (B000202, Invitrogen). Gels were stained with Quick Blue Protein Stain (LU001000, LuBio), and fractions meeting quality criteria were pooled.

## Cells and cell lines

Cell lines were confirmed mycoplasma free by monthly testing using a MycoAlert™ kit (LT07-118, Lonza) following the manufacturer's instructions. *THP1 cells:* THP1 cells (ECACC 88081201, a human monocyte leukemia cell line[51]) were obtained from ECACC and were cultured in T175 flasks at $0.3$-$1 \times 10^6$ cells/mL in Roswell Park Memorial Institute (RPMI) 1640 medium (R0883, Sigma) with 10% FBS (F9665, Sigma) and 5mM L-glutamine (G7513, Sigma), 37 °C, 5% CO2. *J:CXCR1 cells:* The J:CXCR1 cell line was cultured in the same manner as THP1 cells at $0.15$-$1 \times 10^6$ cells per mL[37]. *Activated T-cells:* T-cells were isolated, activated and expanded as follows. Briefly, density gradient centrifugation using Lymphoprep™ density gradient medium (07801, STEMCELL Technologies) and a SepMate™ isolation tube (85450, STEMCELL Technologies) was used to isolate peripheral blood mononuclear cells from human leukocyte cones. Two rounds of purification using a human CD8 + T-cell isolation kit (480011, BioLegend) were used to isolate CD8 + T-cells from peripheral blood mononuclear cells. Activation of CD8 + T-cells was performed in ImmunoCult™-XF T-cell expansion medium (10982, STEMCELL Technologies) with an anti-CD3/anti-CD28 T-cell activator at 25 µl/ml (10991, STEMCELL Technologies), supplemented with 10 ng/ml recombinant human IL-2 (200-02, PeproTech) with 1% penicillin/streptomycin. T-cells were passaged every 2–3 days in TexMACS medium (130-097-196, Miltenyi Biotec), supplemented with 10 ng/ml human IL-2 (200-05-250, Peprotech) and incubated at 37 °C in 5% CO2. CD8 + T-cells were frozen at day 10-13 after initial activation, at $20 \times 10^6$ cell/ml in TexMACS medium supplemented with 10% DMSO (D2650, Sigma) in liquid nitrogen. Activated

CD8+ T-cells were recovered in TexMACs medium at $0.3 \times 10^6$ cell/ml and incubated at 37 °C in 5% CO2, 24 h prior to use. For buffy coat experiments a mixture of primary human monocytes, granulocytes and lymphocytes was isolated from human buffy coat. Briefly, RBCs were sedimented in HetaSep (07806, STEMCELL Technologies) with 1 mL of HetaSep per 5 mL of blood for 15 min at 37 °C. The buffy coat layer was collected from the top and washed in 50 ml MACS buffer (PBS (P4417, Sigma), 0.5% BSA (A7030, Sigma), 0.075% EDTA (E5134, Sigma), pH 7.2), by centrifugation at 2000 g for 10 min at room temperature. The pellet was resuspended in 10 mL of 1x RBC lysis solution (130-094-183, Miltenyi Biotec) and lysed for 10 min at room temperature. The cells were then washed in MACS buffer twice as above. The final pellet was resuspended in 1 mL of MACS buffer per 5 mL of initial blood volume and counted using cell size FSC-H/SSC-H parameters with gate settings previously defined by immunostaining (see below). The leukocytes were then resuspended in assay buffer for cell migration and used within 30 min of isolation.

## Flow cytometry

Cell counts were determined using an ATTUNE NxT Flow Cytometer with Cytkick autosampler (ThermoFisher), based on cell size parameters FSC-H/ SSC-H, with previously defined gate settings[37]. FSC-H/SSC-H gate settings for primary white blood cells from buffy coats were determined using flow cytometry to establish the granulocyte, monocyte and lymphocyte populations (Supplementary Fig. 13). Briefly, cells were labelled with anti-CD3 (Miltenyi Biotec, 130-113-135) to identify the T cell population[86], anti-CD14 (Miltenyi Biotec, 130-110-578) to identify the monocyte population[87], and with anti-CD16 (Miltenyi Biotec, 130-113-951) to identify the granulocyte population[88]. Negative control cells were labelled with anti-human IgG1 (Miltenyi Biotec, 130-113-434). All antibodies were allophycocyanin (APC) conjugated. 100,000 white blood cells were added to Eppendorf tubes and centrifuged at 300 g for 3 min. The supernatant was discarded, and the cells resuspended in 98 µL of MACS buffer (1× PBS, 2 mM EDTA and 0.5% BSA, sterile filtered). 2 µl of antibody was added to the cells. Eppendorf tubes were incubated for 10 min in the dark at 4 °C. Cells were centrifuged at 300 g for 3 min, resuspended in 1 mL of MACS, re-centrifuged at 300 g for 3 min, and resuspended in 400 µL MACS. The fluorescence intensities of the samples were determined using an ATTUNE NxT flow cytometer.

## Cell migration assays

Cell migration assays were performed as follows. *THP1 cell migration.* Briefly, 300000 cells/well were added to the top chamber of a 5-µm 96-well Transwell insert (3387, Corning) in 50 µL of cell migration media (RPMI-1640 (R0883, Sigma), 0.5% FBS (F9665, Sigma), 4mM L-Glutamine (G7513, Sigma), 0.05% DMSO (D4540, Sigma)). The bottom chamber contained 150 µL of migration media with chemokine and peptide. Cells were migrated at 37 °C in 5% CO2 for 4 h. *J:CXCR1 cell migration* was performed exactly as above using a 3-µm Transwell insert (3385, Corning). *Activated T-cell migration.* Briefly, 100000 cells/well were added to the top chamber of a 3-µm Transwell insert (3385, Corning) in 50 µL of migration media (HBSS (14025-092, Life Technologies), 0.1% protease free bovine serum albumin (A7030, Sigma), 0.05% DMSO (D4540, Sigma)). The bottom chamber contained 150 µL of migration media with chemokine and peptide. Cells were migrated at 37 °C in 5% CO2 for 2 h. *Buffy coat migration.* Briefly, a mixture of all cell types at 100000 cells/well were added to the top chamber of a 3-µm Transwell insert (3385, Corning) in 50 µL of migration media (HBSS (14025-092, Life Technologies), 0.1% protease free bovine serum albumin (A7030, Sigma), 0.05% DMSO (D4540, Sigma). The bottom chamber contained 150 µL of migration media with chemokine and peptide. Cells were migrated at 37 °C in 5% CO2 for 2 h. *Analysis of cell migration.* The migration plate was shaken at 850 RPM for 10 min, and media from bottom plate transferred to a round-bottomed 96 well plate (353910, Falcon). Cells were counted by flow cytometry as described above. Experiments were performed at chemokine $EC_{80}$ for individual chemotaxis experiments and at $EC_{50}$ for the pooled chemokine experiments, with three technical replicates, and at least three biological replicates. Statistical significance of differences

between control and test groups was evaluated using a two-sided Dunnett's test with multiple testing correction[89]. Chemokine $EC_{80}$ or chemokine pool $EC_{50}$ values were as determined by performing a chemokine dose-response against the cell type and fitting the dose-response curve with a 3-parameter log-logistic model, where the maximal response is fixed at 100% response. Following identification of the chemokine dose that results in a peak response, higher dose data points that showed a reduction from the peak response were excluded from curve-fitting analyses as this indicates receptor desensitization[90]. Exemplar dose-response curves are shown in Supplementary Figs. 14-16, in which the excluded data points are indicated. Summary statistics of $pEC_{80}$ and $pEC_{80}$ values are provided in Supplementary Table 1. $IC_{50}$ was determined by performing inhibitor dose-response experiments and fitting the dose-response curve with a 3-parameter log-logistic model, fixing the bottom of the curve to zero. Dose-response curves were fitted using the function "drm" in R-package drc[91]. For summary statistics and comparisons, $pEC_{50}$, $pEC_{80}$ and $pIC_{50}$ were calculated as the negative $\log_{10}$ of $EC_{50}$, $EC_{80}$ or $IC_{50}$ expressed in moles/litre.

## Structural analyses

Chemokine models in complex with either a peptide, or cognate receptor or another chemokine were generated using the AlphaFold 3 server[70]. Cognate receptors were identified from the IUPHAR database[72]. No cognate receptor was identified for 5 chemokines (CCL3L1, CCL4L1, CXCL14, CXCL4 and CXCL4L1). AlphaFold 3 models were visualized with PyMOL. Arpeggio[71] was used to extract inter-chain bonds and inter-residue distance information from the highest ranked AlphaFold 3 model. As this configuration did not support aromatic bond kekulization, aromatic-aromatic ring (π-π), and atom-ring interactions (e.g. cation-π, donor-π, carbon-π) were not extracted. In accordance with the terms and conditions of AlphaFold 3 usage[70], no peptide or ligand docking to any AlphaFold 3 model or AlphaFold 3 generated output was performed in this study. The outputs published here are not to be used in any commercial activities, including research on behalf of commercial organizations, are not to be used in any automated system used to perform peptide or ligand docking to the AlphaFold 3 generated outputs such as Glide or AutoDock, and are not to be used to train machine learning models or related technology for biomolecular structure prediction.

## Statistics and reproducibility

Sample size and estimated effect sizes were not formally calculated for cell-based or phage-display experiments and no randomization was performed. Investigators were blinded at analysis to group allocation, but not at data collection. Blinding is not relevant in high-throughput data collection experiments where an automated instrument (e.g. ATTUNE for cell-migration) collects data. No relevant data were excluded. Typically, $n = 3$ technical replicates for each of $n = 3$ biological replicates were performed. Replicates were deemed unsuccessful if the positive control within the experiment did not show the expected response. Cell numbers, numbers of technical and biological replicates are based on previous optimization of such experiments[37,92]. Statistical analyses were performed using the R-base package stats and DescTools. All analyses used data from biologically independent experiments. Statistical tests used were a one-way ANOVA followed by a two-sided Dunnett's post-hoc multiple comparison procedure for comparing several tests with a control[89]. Data are displayed as Tukey box-whisker plots using the ggplot function geom_boxplot, and displays median, lower and upper hinges (25th and 75th percentiles), and whiskers from hinge to 1.5* interquartile range. Exact $P$-values and sample sizes are provided in Supplementary Tables 9a-g.

## Reporting summary

Further information on research design is available in the Nature Portfolio Reporting Summary linked to this article.

## Data availability

The authors declare that the data supporting the findings of this study are available within the paper and its supplementary information files. Plasmid P1909 is deposited with Addgene (ID 233698). Supplementary Figs. and Tables are provided in Supplementary Information. Source data for figures is provided in Supplementary Data 1. Raw data (including DNA sequencing fastq files, and highest ranked model ".cif" files generated by AlphaFold 3), code and analysis outputs are provided in Supplementary Data 2.

## Code availability

Code used is provided with this paper in Supplementary Data 2. Data were analyzed using R version 4.4.0, RStudio version 2024.12.0 + 467, running on aarch64-apple-darwin20 on a Mac Studio with an Apple M1 Ultra Chip running macOS Sonoma 14.5. R packages used were: BiocGenerics_0.48.1, BiocGenerics_0.50.0, Biostrings_2.70.3, Biostrings_2.72.1, dendsort_0.3.4, DescTools_0.99.54, dplyr_1.1.4, drc_3.0-1, flextable_0.9.5, flextable_0.9.6, forcats_1.0.0, GenomeInfoDb_1.38.8, GenomeInfoDb_1.40.0, ggmsa_1.10.0, ggplot2_3.4.4, ggplot2_3.5.1, ggpubr_0.6.0, ggseqlogo_0.2, ggupset_0.3.0, gridExtra_2.3, IRanges_2.36.0, IRanges_2.38.0, janitor_2.2.0, lubridate_1.9.3, MASS_7.3-60.2, msa_1.36.0, officer_0.6.5, officer_0.6.6, pals_1.8, pheatmap_1.0.12, purrr_1.0.2, R.methodsS3_1.8.2, R.oo_1.26.0, R.utils_2.12.3, RColorBrewer_1.1-3, readr_2.1.5, S4Vectors_0.40.2, S4Vectors_0.42.0, scales_1.3.0, strex_2.0.0, stringdist_0.9.12, stringr_1.5.1, tibble_3.2.1, tidyr_1.3.1, tidyverse_2.0.0, writexl_1.5.0, XVector_0.42.0, XVector_0.44.0. R-packages were obtained from the Comprehensive R Archive Network https://cran.r-project.org and Bioconductor (https://www.bioconductor.org). Python packages were obtained from Anaconda (https://anaconda.org). Arpeggio analysis (https://github.com/PDBeurope/arpeggio) used python 3.9.21, biopython 1.84, openbabel 3.1.1, gemmi 0.7 and pdbe-arpeggio 1.4.4[71]. Open-Source PyMOL (https://pymolwiki.org/index.php/MAC_Install) was used for scripting and PyMOL 2.5.2 (https://pymol.org/2/) for visualization of structural models.

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

## Acknowledgements

This research was funded by British Heart Foundation Chair Award (CH/09/003/26631), BHF Program Grant (RG/18/1/33351 and RG/F/23/11021), Oxford BHF Centre of Research Excellence (RE/13/1/30181) and Diabetes-UK/Steve Morgan Foundation Grand Challenge (24/0006744) Awards to SB. MP is funded by a studentship from the Radcliffe Department of Medicine and the Clarendon Scholarship Fund. CC is funded by a studentship from a Wellcome Trust Doctoral Training Program in Genomic Medicine and Statistics.

## Author contributions

Phage-display libraries were designed by SB and constructed by JK. Phage-display screens were performed by JK. Chemotaxis assays were performed by JK, SV, MP, GS, GD and SC. SB conceived and designed the study and performed bioinformatic and statistical analyses. SB and CC performed structural analysis. SB and GD supervised and interpreted experiments. All authors contributed to writing the manuscript.

## Competing interests

The authors declare no competing interests.
