## [Transparent Peer Review file · Communications Biology]

Development of chemokine network inhibitors using combinatorial saturation mutagenesis

Corresponding Author: Professor Shoumo Bhattacharya

This manuscript has been previously reviewed at another journal. This document only contains information relating to versions considered at Communications Biology.

Version 0:

Reviewer comments:

Reviewer #1

(Remarks to the Author)

In the paper "Development of chemokine network inhibitors using combinatorial saturation mutagenesis" by Bhattacharya et al. They developed a method of utilizing phage display assays to identify mutations in chemokine-neutralizing peptides which can enhance and expand their binding promiscuity to multiple chemokines. Using their lead peptide HD2, saturated mutagenesis phage display was used to identify point mutations that improve peptide-chemokine binding. A phage display was then generated to identify mutation combinations that synergies chemokine binding mutations. Inhibitory effects of generated peptides were validated in cells utilizing cell migration assays stimulated by known disease-associated chemokine pools.

This paper presents an innovative method of utilizing phage display to modify pre-existing chemokine inhibitory peptides to enhance their potency and affinity across the whole chemokine family. However, there is an implication made in the paper that broad suppression of chemokine activity only have therapeutic effects, rather than like the parental strain of immunosuppression. Rather, select targeting of chemokines associated with specific diseases may yield more benefits effects. Further development of this methodology will probably be needed to optimize the inhibition of certain chemokines to minimize off-target effects. In summary, this paper present novel ideas of building upon pre-existing anti-inflammatory peptides as broad-spectrum therapeutic purposes against chemokines.

Major

In Figure 2, comparisons between HD2 mutations for binding to all chemokines, CC chemokines and CXnC chemokines has interesting similarities between displayed graph for impacts on mutations for log₂E for all and CC chemokines. CXnC do not display the same pattern. This suggests that the consolidated chemokine phage-display HD2 mutations (Fig 2b) is biased towards CC chemokines, due to CC chemokines having higher representation in the phage display (Fig 2a). An analysis where effects of both CC and CXnC chemokines groups were weighted equally would be helpful in indicating the disproportion representation of CC chemokines do not result in results that favor development of CC chemokines inhibitors. In line 159-163, authors claim that certain mutations (Y5W, T6D, L111 etc) have additive or co-operative enhanced binding to chemokines. The statement is better supported if a figure comparing the individual mutations to the permutation mutants to show a progressive increase in binding as mutations are added.

In line 249-250, authors suggested tryptophan mutations has been shown in their assay to significantly increase their peptide's chemokine binding and inhibitory function. This discussion would be greatly aided if the authors performed structure-function studies comparing Trp with Tyr and Phe at the same positions. Additionally, modifications of amino acids residues such as sulfated tyrosine, which has been known to play a role in enhancing chemokine binding affinity for the chemokine receptors, may be worth mentioning and testing in future experiments.

Minor

Explanation for selection of biotinylated chemokines and non-biotinylated chemokine in the results would aid in understanding why certain chemokines were picked or skipped.

Abbreviations (SCR, ATC, THP1) for Figure 2 and 6 were not defined in figure legend

Figure 3, 6 and 7 Legend (Line 732-733, 766-767, 790-791): - should be = for the p-values.

Figure 7 legend: Abbreviation for RA should be "Rheumatoid arthritis" or "Rheumatoid arthritis synovium" as stated in discussion

p.10 – break up into multiple paragraphs for readability

Reviewer #2

(Remarks to the Author)

In the manuscript "Development of chemokine network inhibitors using combinatorial saturation mutagenesis", the authors developed a combinatorial saturation mutagenesis optimisation strategy (CoSMOS). combination with phage-display, combinatorially mutated evasin-derived peptides with enhanced anti-chemokine potency and breadth of action were successfully identified and characterized. The engineered peptides inhibit the functional activity of pools of chemokines known to be present in atherosclerotic plaque, cytokine-stimulated islets, and rheumatoid arthritis synovial tissues. This is a nicely written paper and the strategy is thoughtful. As someone who has his roots in protein engineering, I think it should get published with minor revisions as noted below:

1) I was intrigued to see the iterative mutagenesis analysis here, please provide a figure or table to represent the enzyme kinetic data (e.g., kcat/km, U, etc) together with the thermostability data (e.g., Tm, etc) of the best mutant generated in each round of mutagenesis.

2) I wonder does the engineered peptides exhibit a general or diverse secondary/tertiary structures? It would be more insightful if the authors make some sequence-structure-function analysis of the improved peptides.

Reviewer #3

(Remarks to the Author)

The manuscript by Kryukova et al. describes an effort to engineer tick-evasin-derived peptides for potency and polypharmacology, as broad-spectrum chemokine inhibitors for inflammatory indications. Via combinatorial saturation mutagenesis and phage display, the authors demonstrate optimization of a previously identified 16-mer peptide HD2 (derived from the N-terminus of EVA4), to improve its inhibitory activity against both CC- and CXC family chemokines.

It is a strong and interesting study. Combined with the authors' earlier work on class A evasin peptide screening, the outcome of this study is a series of evasin-derived peptides with increased effectiveness and potentially decreased immunogenicity, with a potential impact on clinical translatability. The presented approach, CoSMOS, has significant potential in application to other redundant cytokine networks for broad suppression. This said, we have a number of suggestions for improving the manuscript.

Major comments

* The quantitative impact of the optimization efforts seems to be quite modest. IC50 data is presented for only 5 peptides (plus the parent peptide HD2) and in discrepant assays (T12W only compared to HD2 and only in THP1 migration towards CCL5; CM304 only to Y5W and only in ATC migration towards CXCL12, etc.). The presented IC50s are mostly in low micromolar range (i.e. not very potent) and the presented IC50 improvements only in a 2-3 fold range. Dose response curves are not shown. We suspect the biggest impact of optimization was in broadening the range of inhibited chemokines, but this is not very well articulated or quantified. Recommend to select 1-2 most potent analogs and systematically quantify its potency and selectivity in comparison with the parent peptide.

* In addition to peptide variants with improved binding to both families (e.g. CM307 and CM539), the authors identified peptide variants that improved chemokine binding in a subfamily-selective manner: e.g. T6D, L11I, T12W, CM304 and CM452, for the CC chemokine family; A7D, L11I, T16D, and CM526 for CXnC chemokine family. These variants were then used in migrations assays on three cell lines (THP1, activated T-cells, and CXCR1-expressing Jurkat cells) towards ten (Fig. 3) or nine (Fig. 5) chemokines. Unfortunately, the multidimensional nature of the obtained data is practically lost in Figs. 3 and 6 (all three cell lines are folded together) and no analysis is presented as to which inhibitory profile results in better inhibition of which cell line towards which chemokine. This should be analyzed and explained in Results.

* "The most effective individual mutations with significant improvement over HD2 in this analysis were Y5W, T6W, A9W and T16C": three mutations to a Trp (which likely makes the peptides less soluble) and one to a Cys (which likely causes the peptides to dimerize or bind covalently). Top selected peptides need to be assessed for changes in solubility and stoichiometry.

* A big miss in our view is not attempting to draw or rationalize the structure-activity relationships. Considering that structures of evasins with chemokines are available, modeling could be quite reliable and would go a long way towards the understanding of molecular mechanisms that govern broad anti-chemokine activity of the identified peptides.

* Figure 7 shows a number of intriguing patterns. For the ATHERO chemokine mixture, the migration of ATC but not monocytes were inhibited by practically all peptides. For the INS mixture, similarly, ATC migration is practically abrogated by about half of the peptides. By contrast, for the RA chemokine mixture, the migration of ATC seems to be enhanced by all peptides including the scrambled control. Is it possible to deconvolute these patterns considering the known chemokine mixture compositions and peptide selectivity preferences?

* Chemokines are said to be used at EC80 doses but this is unclear as the migration dose response curves are usually not

sigmoidal. This needs to be shown and clarified via a supplemental figure.

Minor comments

* Abstract: "Tick evasins overcome chemokine networks" is a misleading statement, it would be better to explain it in terms of "broad specificity" or "multi-targeted effects" of tick evasins. Same in lines 47 and 49 in the Introduction.

* Introduction, Line 31: a space is needed in "through18"

* Phage-display selection was performed against 22 biotinylated chemokines but in the followup migration experiments, "seven CC and three CXC chemokines" were chosen, without any rationale for doing so. Similarly, "THP1 cells, activated CD8+T-cells (ATC), and Jurkat cells expressing CXCR1" were chosen as cell lines but no rationale for this selection was provided. Some immunological background info about the selected cells and chemokines, and a rationale for their use in migration experiments, would be very helpful.

* Figs. 3a and 6a are not reader-friendly: not only the nature of the tested cell lines is omitted, but also the gray background and the selected color rainbow for chemokines makes them hardly discernible.

* Fig. 3b-c, 6b-c, and similar plots in Supplemental Figures: the dot matrices are space-consuming and somewhat redundant. Cell line and chemokine could be given in the plot header, and peptide mutants as labels on the x-axis

* In Fig 5a, there is a mismatch between "CM-maxCXnC" colors in the heatmap and the legend.

* For explicitness and transparency, instead of (or in addition to) Figs 3(b-c) and 6(b-c), we strongly recommend showing peptide dose response curves and with individual biological replicates as dots.

* A schematic of the phagemid design, and a schematic of the parallel assay with multiple immobilized chemokines, are needed.

* Results Line 100: "...log₂E, which is correlated with binding affinity [42]" should be "inversely correlated" (or otherwise clarify the relationship).

* Results Line 128: We believe both Fig 3b and c need to be called out.

* Results Line 128: "We also explored the potency of HD2 mutant peptides against CC and CXC chemokines using dose-response assays" - we could not find dose response data for any CC or CXC chemokines except CCL5 and CXCL10; recommend to change wording to "... against CCL5 and CXCL10..."

Reviewed by:
Irina Kufareva, PhD
Kai Wang, PhD

Reviewer #4

(Remarks to the Author)

Version 2:

Reviewer comments:

Reviewer #2

(Remarks to the Author)

The authors did extensive revisions for the original manuscript. I recommend to accept it in the current form.

Reviewer #3

(Remarks to the Author)

This is a revised version of the manuscript by Kryukova et al. describing an effort to engineer tick-evasin-derived peptides for potency and polypharmacology, as broad-spectrum chemokine inhibitors for inflammatory indications. In the revision, the authors adequately addressed most of our critiques, most by including the requested data and a few by providing a well-justified rebuttal. The addition of the structural models and an ensuing structure-activity relationship analysis greatly strengthened the manuscript in our view. The inclusion of complete analysis code and models into the paper's supplementary materials is also commendable as it helps hold high standards of rigor & reproducibility.

We only have two minor recommendations:

* Please confirm that Arpeggio and other analyzes in the manuscript do not violate the Terms of Use

<https://alphafoldserver.com/terms> (“You must not use AlphaFold Server or its outputs in any automated system that predicts the binding or interaction of the protein with ligands or peptides”)

* The use of “negative” in relation to the controls in Supp. Tables 1b-1f is confusing as the controls are sometimes negative and sometimes positive. Recommend rephrasing simply as “comparisons to control”, especially since the control is explicitly specified in each table.

Reviewed by:
Irina Kufareva, PhD
Kai Wang, PhD

Reviewer #4

(Remarks to the Author)

This is a revised version of the manuscript by Kryukova et al. describing an effort to engineer tick-evasin-derived peptides for potency and polypharmacology, as broad-spectrum chemokine inhibitors for inflammatory indications. In the revision, the authors adequately addressed most of our critiques, most by including the requested data and a few by providing a well-justified rebuttal. The addition of the structural models and an ensuing structure-activity relationship analysis greatly strengthened the manuscript in our view. The inclusion of complete analysis code and models into the paper's supplementary materials is also commendable as it helps hold high standards of rigor & reproducibility.

We only have two minor recommendations:

Please confirm that Arpeggio and other analyzes in the manuscript do not violate the Terms of Use

<https://alphafoldserver.com/terms> (“You must not use AlphaFold Server or its outputs in any automated system that predicts the binding or interaction of the protein with ligands or peptides”)

The use of “negative” in relation to the controls in Supp. Tables 1b-1f is confusing as the controls are sometimes negative and sometimes positive. Recommend rephrasing simply as “comparisons to control”, especially since the control is explicitly specified in each table.

Reviewed by:
Irina Kufareva, PhD
Kai Wang, PhD
